# Shadowing Properties of Optimization Algorithms

**Antonio Orvieto**
Department of Computer Science
ETH Zurich, Switzerland *

**Aurelien Lucchi**
Department of Computer Science
ETH Zurich, Switzerland

## Abstract

Ordinary differential equation (ODE) models of gradient-based optimization methods can provide insights into the dynamics of learning and inspire the design of new algorithms. Unfortunately, this thought-provoking perspective is weakened by the fact that — in the worst case — the error between the algorithm steps and its ODE approximation grows exponentially with the number of iterations. In an attempt to encourage the use of continuous-time methods in optimization, we show that, if some additional regularity on the objective is assumed, the ODE representations of Gradient Descent and Heavy-ball do not suffer from the aforementioned problem, once we allow for a small perturbation on the algorithm initial condition. In the dynamical systems literature, this phenomenon is called *shadowing*. Our analysis relies on the concept of *hyperbolicity*, as well as on tools from numerical analysis.

## 1 Introduction

We consider the problem of minimizing a smooth function $f : \mathbb{R}^d \to \mathbb{R}$. This is commonly solved using gradient-based algorithms due to their simplicity and provable convergence guarantees. Two of these approaches — that prevail in machine learning — are: 1) Gradient Descent (GD), that computes a sequence $(x_k)_{k=0}^{\infty}$ of approximations to the minimizer recursively:

$$x_{k+1} = x_k - \eta \nabla f(x_k), \tag{GD}$$

given an initial point $x_0$ and a *learning rate* $\eta > 0$; and 2) Heavy-ball (HB)[43] that computes a different sequence of approximations $(z_k)_{k=0}^{\infty}$ such that

$$z_{k+1} = z_k + \beta(z_k - z_{k-1}) - \eta \nabla f(z_k), \tag{HB}$$

given an initial point $z_0 = z_{-1}$ and a *momentum* $\beta \in [0, 1)$. A method related to HB is Nesterov's accelerated gradient (NAG), for which $\nabla f$ is evaluated at a different point (see Eq. 2.2.22 in [38]).

Analyzing the convergence properties of these algorithms can be complex, especially for NAG whose convergence proof relies on algebraic tricks that reveal little detail about the acceleration phenomenon, i.e. the celebrated optimality of NAG in convex smooth optimization. Instead, an alternative approach is to view these methods as numerical integrators of some ordinary differential equations (ODEs). For instance, GD performs the explicit Euler method on $\dot{y} = -\nabla f(y)$ and HB the semi-implicit Euler method on $\ddot{q} + \alpha \dot{q} + \nabla f(q) = 0$ [43, 47]. This connection goes beyond the study of the dynamics of learning algorithms, and has recently been used to get a useful and thought-provoking viewpoint on the computations performed by residual neural networks [10, 12].

In optimization, the relation between discrete algorithms and ordinary differential equations is *not new at all*: the first contribution in this direction dates back to, at least, 1958 [18]. This connection has recently been revived by the work of [49, 28], where the authors show that the continuous-time limit of NAG is a second order differential equation with vanishing viscosity. This approach provides an interesting perspective on the somewhat mysterious acceleration phenomenon, connecting it to

the theory of damped nonlinear oscillators and to Bessel functions. Based on the prior work of [52, 27] and follow-up works, it has become clear that the analysis of ODE models can provide simple intuitive proofs of (known) convergence rates and can also lead to the design of new discrete algorithms [55, 4, 53, 54]. Hence, one area of particular interest has been to study the relation between continuous-time models and their discrete analogs, specifically understanding the error resulting from the discretization of an ODE. The conceptual challenge is that, in the worst case, the approximation error of any numerical integrator *grows exponentially*[2] as a function of the integration interval [11, 21]; therefore, convergence rates derived for ODEs *can not* be straightforwardly translated to the corresponding algorithms. In particular, obtaining convergence guarantees for the discrete case requires analyzing a discrete-time Lyapunov function, which often cannot be easily recovered from the one used for the continuous-time analysis [46, 47]. Alternatively, (sophisticated) numerical arguments can be used to get a rate through an approximation of such Lyapunov function [55].

In this work, we follow a different approach and directly study conditions under which the flow of an ODE model is *shadowed* by (i.e. is uniformly close[3] to) the iterates of an optimization algorithm. The key difference with previous work, which makes our analysis possible, is that we allow the algorithm — i.e. the *shadow*— to start from a slightly perturbed initial condition compared to the ODE (see Fig. 2 for an illustration of this point). We rely on tools from numerical analysis [21] as well as concepts from dynamical systems [7], where solutions to ODEs and iterations of algorithm are viewed as the same object, namely maps in a topological space [7]. Specifically, our analysis builds on the theory of hyperbolic sets, which grew out of the works of Anosov [1] and Smale [48] in the 1960's and plays a fundamental role in several branches of the area of dynamical systems but has not yet been seen to have a relationship with optimization for machine learning.

In this work we pioneer the use of the theory of shadowing in optimization. In particular, we show that, if the objective is strongly-convex or if we are close to an hyperbolic saddle point, GD and HB are a shadow of (i.e. follow closely) the corresponding ODE models. We back-up and complement our theory with experiments on machine learning problems.

To the best of our knowledge, our work is the first to focus on a (Lyapunov function independent) systematic and quantitative comparison of ODEs and algorithms for optimization. Also, we believe the tools we describe in this work can be used to advance our understanding of related machine learning problems, perhaps to better characterize the attractors of neural ordinary differential equations [10].

## 2  Background

This section provides a comprehensive overview of some fundamental concepts in the theory of dynamical systems, which we will use heavily in the rest of the paper.

### 2.1  Differential equations and flows

Consider the autonomous differential equation $\dot{y} = g(y)$. Every $y$ represents a point in $\mathbb{R}^n$ (a.k.a phase space) and $g : \mathbb{R}^n \to \mathbb{R}^n$ is a vector field which, at any point, prescribes the velocity of the solution $y$ that passes through that point. Formally, the curve $y : \mathbb{R} \to \mathbb{R}^n$ is a solution passing through $y_0$ at time 0 if $\dot{y}(t) = g(y(t))$ for $t \in \mathbb{R}$ and $y(0) = y_0$. We call this the solution to the initial value problem (IVP) associated with $g$ (starting at $y_0$). The following results can be found in [42, 26].

**Theorem 1.** *Assume $g$ is Lipschitz continuous and $C^k$. The IVP has a unique $C^{k+1}$ solution in $\mathbb{R}$.*

This fundamental theorem tells us that, if we integrate for $t$ time units from position $y_0$, the final position $y(t)$ is uniquely determined. Therefore, we can define the family $\{\varphi_t^g\}_{t \in \mathbb{R}}$ of maps — the **flow** of $g$ — such that $\varphi_t^g(y_0)$ is the solution at time $t$ of the IVP. Intuitively, we can think of $\varphi_t^g(y_0)$ as determining the location of a particle starting at $y_0$ and moving via the velocity field $g$ for $t$ seconds. Since the vector field is static, we can move along the solution (in both directions) by iteratively applying this map (or its inverse). This is formalized below.

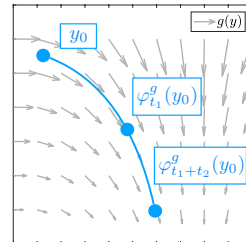

**Proposition 1.** *Assume $g$ is Lipschitz continuous and $C^k$. For any $t \in \mathbb{R}$, $\varphi_t^g \in C^{k+1}$ and, for any $t_1, t_2 \in \mathbb{R}$, $\varphi_{t_1+t_2}^g = \varphi_{t_1}^g \circ \varphi_{t_2}^g$. In particular, $\varphi_t^g$ is a diffeomorphism[4] with inverse $\varphi_{-t}^g$.*

In a way, the flow allows us to *exactly discretize* the trajectory of an ODE. Indeed, let us fix a stepsize $h > 0$; the associated **time-h map** $\varphi_h^g$ integrates the ODE $\dot{y} = g(y)$ for $h$ seconds starting from some $y_0$, and we can apply this map recursively to compute a sampled ODE solution. In this paper, we study how flows can approximate the iterations of some optimization algorithm using the language of dynamical systems and the concept of shadowing.

## 2.2 Dynamical systems and shadowing

A dynamical system on $\mathbb{R}^n$ is a map $\Psi : \mathbb{R}^n \to \mathbb{R}^n$. For $p \in \mathbb{N}$, we define $\Psi^p = \Psi \circ \cdots \circ \Psi$ ($p$ times). If $\Psi$ is invertible, then $\Psi^{-p} = \Psi^{-1} \circ \cdots \circ \Psi^{-1}$ ($p$ times). Since $\Psi^{p+m} = \Psi^p \circ \Psi^m$, these iterates form a group if $\Psi$ is invertible, and a semigroup otherwise. We proceed with more definitions.

**Definition 1.** *A sequence $(x_k)_{k=0}^\infty$ is a (positive) **orbit** of $\Psi$ if, for all $k \in \mathbb{N}$, $x_{k+1} = \Psi(x_k)$.*

For the rest of this subsection, the reader may think of $\Psi$ as an optimization algorithm (such as GD, which maps $x$ to $x - \eta \nabla f(x)$ ) and of the orbit $(x_k)_{k=0}^\infty$ as its iterates. Also, the reader may think of $(y_k)_{k=0}^\infty$ as the sequence of points derived from the iterative application of $\varphi_h^g$, which is itself a dynamical system, from some $y_0$. The latter sequence represents our ODE approximation of the algorithm $\Psi$. Our goal in this subsection is to understand when a sequence $(y_k)_{k=0}^\infty$ is "close to" an orbit of $\Psi$. The first notion of such similarity is local.

**Definition 2.** *The sequence $(y_k)_{k=0}^\infty$ is a $\delta$−**pseudo-orbit** of $\Psi$ if, for all $k \in \mathbb{N}$, $\|y_{k+1} - \Psi(y_k)\| \leq \delta$.*

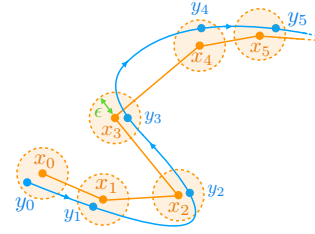

If $(y_k)_{k=0}^\infty$ is locally similar to an orbit of $\Psi$ (i.e. it is a pseudo-orbit of $\Psi$), then we may hope that such similarity extends globally. This is captured by the concept of shadowing.

**Definition 3.** *A pseudo-orbit $(y_k)_{k=0}^\infty$ of $\Psi$ is $\epsilon$−**shadowed** if there exists an orbit $(x_k)_{k=0}^\infty$ of $\Psi$ such that, for all $k \in \mathbb{N}$, $\|x_k - y_k\| \leq \epsilon$.*

It is crucial to notice that, as depicted in the figure above, we allow the shadowing orbit (a.k.a. the shadow) to start at a *perturbed point* $x_0 \neq y_0$. A natural question is the following: *which properties must $\Psi$ have such that a general pseudo-orbit is shadowed?* A lot of research has been carried out in the last decades on this topic (see e.g. [41, 31] for a comprehensive survey).

**Shadowing for contractive/expanding maps.** A straight-forward sufficient condition is related to contraction. $\Psi$ is said to be *uniformly contracting* if there exists $\rho < 1$ (*contraction factor*) such that for all $x_1, x_2 \in \mathbb{R}^n$, $\|\Psi(x_1) - \Psi(x_2)\| \leq \rho\|x_1 - x_2\|$. The next result can be found in [23].

**Theorem 2.** *(Contraction map shadowing theorem) Assume $\Psi$ is uniformly contracting. For every $\epsilon > 0$ there exists $\delta > 0$ such that every $\delta$−pseudo-orbit $(y_k)_{k=0}^\infty$ of $\Psi$ is $\epsilon$−shadowed by the orbit $(x_k)_{k=0}^\infty$ of $\Psi$ starting at $x_0 = y_0$, that is $x_k := \Psi^k(x_0)$. Moreover, $\delta \leq (1 - \rho)\epsilon$.*

The idea behind this result is simple: since all distances are contracted, errors that are made along the pseudo-orbit vanish asymptotically. For instructive purposes, we report the full proof.

*Proof:* We proceed by induction: the proposition is trivially true at $k = 0$, since $\|x_0 - y_0\| \leq \epsilon$; next, we assume the proposition holds at $k \in \mathbb{N}$ and we show validity for $k + 1$. We have

$$
\begin{aligned}
\|x_{k+1} - y_{k+1}\| &\overset{\text{subadditivity}}{\leq} \|\Psi(x_k) - \Psi(y_k)\| + \|\Psi(y_k) - y_{k+1}\| \\
&\overset{\delta\text{-pseudo-orbit}}{\leq} \|\Psi(x_k) - \Psi(y_k)\| + \delta \\
&\overset{\text{contraction}}{\leq} \rho\|x_k - y_k\| + \delta \\
&\overset{\text{induction}}{\leq} \rho\epsilon + \delta.
\end{aligned}
$$

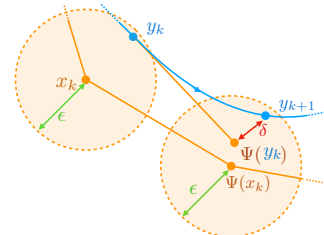

Finally, since $\delta \leq \epsilon(1-\rho)$, $\rho\epsilon + \delta = \epsilon$. ∎

Next, assume $\Psi$ is invertible. If $\Psi$ is **uniformly expanding** (i.e. $\rho > 1$), then $\Psi^{-1}$ is uniformly contracting with contraction factor $1/\rho$ and we can apply the same reasoning backwards: consider the pseudo-orbit $\{y_0, y_1, \cdots, y_K\}$ up to iteration $K$, and set $x_K = y_K$ (before, we had $x_0 = y_0$); then, apply the same reasoning as above using the map $\Psi^{-1}$ and find the initial condition $x_0 = \Psi^{-K}(y_K)$. In App. B.2 we show that this reasoning holds in the limit, i.e. that $\Psi^{-K}(y_K)$ converges to a suitable $x_0$ to construct a shadowing orbit under $\delta \leq (1 - 1/\rho)\epsilon$ (precise statement in Thm. B.3).

**Shadowing in hyperbolic sets.** In general, for machine learning problems, the algorithm map $\Psi$ can be a combination of the two cases above: an example is the pathological contraction-expansion behavior of Gradient Descent around a saddle point [16]. To illustrate the shadowing properties of such systems, we shall start by taking $\Psi$ to be linear[5], that is $\Psi(x) = Ax$ for some $A \in \mathbb{R}^{n \times n}$. $\Psi$ is called **linear hyperbolic** if its spectrum $\sigma(A)$ does not intersect the unit circle $\mathbb{S}^1$. If this is the case, we call $\sigma_s(A)$ the part of $\sigma(A)$ inside $\mathbb{S}^1$ and $\sigma_u(A)$ the part of $\sigma(A)$ outside $\mathbb{S}^1$. The spectral decomposition theorem (see e.g. Corollary 4.56 in [24]) ensures that $\mathbb{R}^n$ decomposes into two $\Psi$-invariant closed subspaces (a.k.a the stable and unstable manifolds) in direct sum : $\mathbb{R}^n = E_s \oplus E_u$. We call $\Psi_s$ and $\Psi_u$ the restrictions of $\Psi$ to these subspaces and $A_s$ and $A_u$ the corresponding matrix representations; the theorem also ensures that $\sigma(A_s) = \sigma_s(A)$ and $\sigma(A_u) = \sigma_u(A)$. Moreover, using standard results in spectral theory (see e.g. App. 4 in [24]) we can equip $E_s$ and $E_u$ with norms equivalent to the standard Euclidean norm $\|\cdot\|$ such that, w.r.t. the new norms, $\Psi_u$ is uniformly expanding and $\Psi_s$ uniformly contracting. If $A$ is symmetric, then it is diagonalizable and the norms above can be taken to be Euclidean. To wrap up this paragraph — we can think of a linear hyperbolic system as a map that allows us to decouple its stable and unstable components *consistently* across $\mathbb{R}^n$. Therefore, a shadowing result directly follows from a combination of Thm. 2 and B.3 [23].

An important further question is — whether the result above for linear maps can be generalized. From the classic theory of nonlinear systems [26, 2], we know that in a neighborhood of an **hyperbolic point** p for $\Psi$, that is $D\Psi(p)$ has no eigenvalues on $\mathbb{S}^1$, $\Psi$ behaves like[6] a linear system. Similarly, in the analysis of optimization algorithms, it is often used that, near a saddle point, an Hessian-smooth function can be approximated by a quadratic [35, 15, 19]. Hence, it should not be surprising that pseudo-orbits of $\Psi$ are shadowed in a neighborhood of an hyperbolic point, if such set is $\Psi$-invariant [31]: this happens for instance if $p$ is a stable equilibrium point (see definition in [26]).

Starting from the reasoning above, the celebrated *shadowing theorem*, which has its foundations in the work of Ansosov [1] and was originally proved in [6], provides the strongest known result of this line of research: near an **hyperbolic set** of $\Psi$, pseudo-orbits are shadowed. Informally, $\Lambda \subset \mathbb{R}^n$ is called hyperbolic if it is $\Psi$-invariant and has clearly marked local directions of expansion and contraction which make $\Psi$ behave similarly to a linear hyperbolic system. We provide the precise definition of hyperbolic set and the statement of the shadowing theorem in App. B.1.

Unfortunately, despite the ubiquity of hyperbolic sets, it is practically infeasible to establish this property analytically [3]. Therefore, an important part of the literature [14, 44, 11, 51] is concerned with the numerical verification of the existence of a shadowing orbit *a posteriori*, i.e. given a particular pseudo-orbit.

## 3 The Gradient Descent ODE

We assume some regularity on the objective function $f : \mathbb{R}^d \to \mathbb{R}$ which we seek to minimize.

**(H1)** $f \in C^2(\mathbb{R}^d, \mathbb{R})$ is coercive[7], bounded from below and $L$-smooth ($\forall a \in \mathbb{R}^d, \|\nabla^2 f(a)\| \leq L$).

In this chapter, we study the well-known continuous-time model for GD : $\dot{y} = \nabla f(y)$ (GD-ODE). Under **(H1)**, Thm. 1 ensures that the solution to GD-ODE exists and is unique. We denote by $\varphi_h^{\text{GD}}$ the corresponding time-$h$ map. We show that, under some additional assumptions, the orbit of $\varphi_h^{\text{GD}}$ (which we indicate as $(y_k)_{k=0}^{\infty}$) is shadowed by an orbit of the GD map with learning rate $h$: $\Psi_h^{\text{GD}}$.

**Remark 1** (Bound on the ODE gradients). *Under **(H1)** let $G_y = \{p : p = \|\nabla f(y(t))\|, t \geq 0\}$ be the set of gradient magnitudes experienced along the GD-ODE solution starting at any $y_0$. It is easy to prove, using an argument similar to Prop. 2.2 in [8], that coercivity implies $\sup G_y < \infty$. A similar argument holds for the iterates of Gradient Descent. Hence, for the rest of this chapter it is safe to assume that gradients are bounded: $\|\nabla f(y(t))\| \leq \ell$ for all $t \geq 0$. For instance, if $f$ is a quadratic centered at $x^*$, then we have $\ell = L\|y_0 - x^*\|$.*

The next result follows from the fact that GD implements the explicit Euler method on GD-ODE.

**Proposition 2.** *Assume **(H1)**. $(y_k)_{k=0}^\infty$ is a $\delta$-pseudo-orbit of $\Psi_h^{GD}$ with $\delta = \frac{\ell L}{2}h^2$: $\forall k \in \mathbb{N}$,*

$$\|y_{k+1} - \Psi_h^{GD}(y_k)\| \leq \delta.$$

*Proof.* Thanks to Thm. 1, since the solution $y$ of GD-ODE is a $C^2$ curve, we can write $y(kh + h) = \varphi_h^{GD}(y(kh)) = y_{k+1}$ using Taylor's formula with Lagrange's Remainder in Banach spaces (see e.g. Thm 5.2. in [13]) around time $t = kh$. Namely: $y(kh + h) = y(kh) - \dot{y}(kh)h + \mathcal{R}(2, h)$, where $\mathcal{R}(2, \cdot) : \mathbb{R}_{>0} \to \mathbb{R}^d$ is the approximation error as a function of $h$, which can be bounded as follows:

$$\|\mathcal{R}(2, h)\| \leq \frac{h^2}{2} \sup_{0 \leq \lambda \leq 1} \|\ddot{y}(t + \lambda h)\| = \frac{h^2}{2} \sup_{0 \leq \lambda \leq 1} \|\nabla^2 f(y(t + \lambda h))\nabla f(y(t + \lambda h))\| \leq$$

$$\leq \frac{h^2}{2} \sup_{0 \leq \lambda \leq 1} \|\nabla^2 f(y(t + \lambda h))\| \sup_{0 \leq \lambda \leq 1} \|\nabla f(y(t + \lambda h))\| \leq \frac{h^2}{2} L\ell.$$

Hence, since $y(kh) - \dot{y}(kh)h = \Psi_h^{GD}(y_k)$, we have $\|y_{k+1} - \Psi_h^{GD}(y_k)\| \leq \frac{\ell L}{2}h^2$. ∎

## 3.1 Shadowing under strong convexity

As seen in Sec. 2.2, the last proposition provides the first step towards a shadowing result. We also discussed that if, in addition, $\Psi_h^{GD}$ is a contraction, we directly have shadowing (Thm. 1). Therefore, we start with the following assumption that will be shown to imply contraction.

**(H2)** $f$ is $\mu$-strongly-convex: for all $a \in \mathbb{R}^d$, $\|\nabla^2 f(a)\| \geq \mu$.

The next result follows from standard techniques in convex optimization (see e.g. [25]).

**Proposition 3.** *Assume **(H1)**, **(H2)**. If $0 < h \leq \frac{1}{L}$, $\Psi_h^{GD}$ is uniformly contracting with $\rho = 1 - h\mu$.*

We provide the proof in App. D.1 and sketch the idea using a quadratic form: let $f(x) = \langle x, Hx \rangle$ with $H$ symmetric s.t. $\mu I \preceq H \preceq LI$; if $h \leq \frac{1}{L}$ then $(1 - Lh) \leq \|I - hH\| \leq (1 - \mu h)$. Prop. 3 follows directly: $\|\Psi_h^{GD}(x) - \Psi_h^{GD}(y)\| = \|(I - hH)(x - y)\| \leq \|I - hH\|\|x - y\| \leq \rho\|x - y\|$.

The shadowing result for strongly-convex functions is then a simple application of Thm. 2.

**Theorem 3.** *Assume **(H1)**, **(H2)** and let $\epsilon$ be the desired accuracy. Fix $0 < h \leq \min\{\frac{2\mu\epsilon}{L\ell}, \frac{1}{L}\}$; the orbit $(y_k)_{k=0}^\infty$ of $\varphi_h^{GD}$ is $\epsilon$-shadowed by any orbit $(x_k)_{k=0}^\infty$ of $\Psi_h^{GD}$ with $x_0$ such that $\|x_0 - y_0\| \leq \epsilon$.*

*Proof.* From Thm. 2, we need $(y_k)_{k=0}^\infty$ to be a $\delta$-pseudo-orbit of $\Psi_h^{GD}$ with $\delta \leq (1 - \rho)\epsilon$. From Prop. 2 we know $\delta = \frac{\ell L}{2}h^2$, while from Prop. 3 we have $\rho \leq (1 - h\mu)$. Putting it all together, we get $\frac{\ell L}{2}h^2 \leq h\mu\epsilon$, which holds if and only if $h \leq \frac{2\mu\epsilon}{L\ell}$. ∎

Notice that we can formulate the theorem in a dual way: namely, *for every learning rate* we can bound the ODE approximation error (i.e. find the shadowing radius).

**Corollary 1.** *Assume **(H1)**, **(H2)**. If $0 < h \leq \frac{1}{L}$, $(y_k)_{k=0}^\infty$ is $\epsilon$-shadowed by any orbit $(x_k)_{k=0}^\infty$ of $\Psi_h^{GD}$ starting at $x_0$ with $\|x_0 - y_0\| \leq \epsilon$, with $\epsilon = \frac{h\ell L}{2\mu}$.*

This result ensures that if the objective is smooth and strongly-convex, then GD-ODE is a theoretically sound approximation of GD. We validate this in Fig. 1 by integrating GD-ODE analytically.

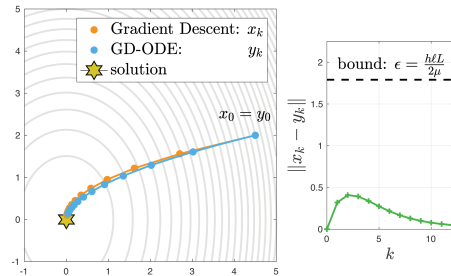

Figure 1: Orbit of $\Psi_h^{GD}$, $\varphi_h^{GD}$ (sampled ODE sol.) on strongly-convex quadratic. $h = 0.2$.

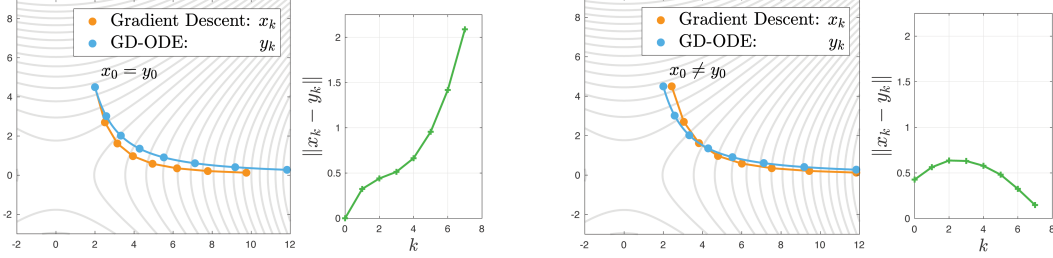

(a) The orbit of GD is *not* a shadow: error blows up.     (b) The orbit of GD is a shadow: error is bounded.

Figure 2: A few iterations of the maps $\Psi_h^{\text{GD}}$ and $\varphi_h^{\text{GD}}$ with different initializations on a quadratic saddle. GD-ODE was solved analytically and $h = 0.2$. On the right plot, the coordinates of $x_0$ are $x_{0,1} = \Psi^{-7}(y_{7,1})$ (expanding direction, need to reverse time) and $x_{0,2} = y_{0,2}$ (contracting direction).

**Sharpness of the result.**    First, we note that the bound for $\delta$ in Prop. 2 cannot be improved; indeed it coincides with the well-known *local* truncation error of Euler's method [22]. Next, pick $f(x) = x^2/2$, $x_0 = 1$ and $h = 1/L = 1$. For $k \in \mathbb{N}$, gradients are smaller than 1 for both GD-ODE and GD, hence $\ell = L = \mu = 1$. Our formula for the *global* shadowing radius gives $\epsilon = hL\ell/(2\mu) = 0.5$, equal to the local error $\delta = \ell L h^2/2$ — i.e. as tight the well-established local result. In fact, GD jumps to 0 in one iteration, while $y(t) = e^{-t}$; hence $y(1) - x_1 = 1/e \approx 0.37 < 0.5$. For smaller steps, like $h = 0.1$, our formula predicts $\epsilon = 0.05 = 10\delta$. In simulation, we have maximum deviation at $k = 10$ and is around $0.02 = 4\delta$, only 2.5 times smaller than our prediction.

**The convex case.**    If $f$ is convex but not strongly-convex, GD is non-expanding and the error between $x_k$ and $y_k$ cannot be bounded by a constant[8] but grows slowly : in App. C.1 we show the error $\epsilon_k$ it is upper bounded by $\delta k = \ell L k h^2/2 = \mathcal{O}(kh^2)$.

**Extension to stochastic gradients.**    We extend Thm. 3 to account for perturbations: let $\Psi_h^{\text{SGD}}(x) = x - h\tilde{\nabla}f(x)$, where $\tilde{\nabla}f(x)$ is a stochastic gradient s.t. $\|\tilde{\nabla}f(x) - \nabla f(x)\| \leq R$. Then, for $\epsilon$ big enough, we can (proof in App. C.2) choose $h \leq \frac{2(\mu\epsilon - R)}{\ell L}$ so that the orbit of $\varphi_h^{\text{GD}}$ (deterministic) is shadowed by the stochastic orbit of $\Psi_h^{\text{SGD}}(x)$ starting from $x_0 = y_0$. So, if the $h$ is small enough, GD-ODE can be used to study SGD. This result is well known from stochastic approximation [30].

### 3.2   Towards non-convexity: behaviour around local maxima and saddles

We first study the strong-concavity case and then combine it with strong-convexity to assess the shadowing properties of GD around a saddle.

**Strong-concavity.**    In this case, it follows from the same argument of Prop. 3 that GD is uniformly expanding with $\rho = 1 + \gamma h > 1$, with $-\gamma := \max(\sigma(H)) < 0$. As mentioned in the background section (see Thm. B.3 for the precise statement) this case is conceptually identical to strong-convexity once we reverse the arrow of time (so that expansions become contractions). We are allowed to make this step because, under **(H1)** and if $h \leq \frac{1}{L}$, $\Psi_h^{\text{GD}}$ is a diffeomorphism (see e.g. [34], Prop. 4.5). In particular, the backwards GD map $(\Psi_h^{\text{GD}})^{-1}$ is contracting with factor $1/\rho$. Consider now the *initial part* of an orbit of GD-ODE such that the gradient norms are still bounded by $\ell$ and let $y_K = (\varphi_h^{\text{GD}})^K(y_0)$ be the last point of such orbit. It is easy to realize that $(y_k)_{k=0}^K$ is a pseudo-orbit, with reversed arrow of time, of $(\Psi_h^{\text{GD}})^{-1}$. Hence, Thm. 3 directly ensures shadowing choosing $x_K = y_K$ and $x_k = (\Psi_h^{\text{GD}})^{k-K}(y_K)$. Crucially — the initial condition of the shadow $(x_k)_{k=0}^K$ we found are slightly[9] *perturbed*: $x_0 = (\Psi_h^{\text{GD}})^{-K}(y_K) \cong y_0$. Notice that, if we instead start GD from exactly $x_0 = y_0$, the iterates will diverge from the ODE trajectory, since every error made along the pseudo-orbit is amplified. We show this for the unstable direction of a saddle in Fig. 2a.

**Quadratic saddles.**    As discussed in Sec. 2, if the space can be split into stable (contracting) and unstable (expanding) invariant subspaces ($\mathbb{R}^d = E_s \oplus E_u$), then every pseudo-orbit is shadowed. This is a particular case of the shadowing theorem for hyperbolic sets [6]. In particular, we saw that

if $\Psi_h^{\mathrm{GD}}$ is linear hyperbolic such splitting exists and $E_s$ and $E_u$ are the subspaces spanned by the stable and unstable eigenvalues, respectively. It is easy to realize that $\Psi_h^{\mathrm{GD}}$ is linear if the objective is a quadratic; indeed $f(x) = \langle x, Hx \rangle$ is such that $\Psi_h^{\mathrm{GD}}(x) = (I - hH)x$. It is essential to note that hyperbolicity requires $H$ to have no eigenvalue at $0$ — i.e. that the saddle has only directions of strictly positive or strictly negative curvature. This splitting allows to study shadowing on $E_s$ and $E_u$ separately: for $E_s$ we can use the shadowing result for strong-convexity and for $E_u$ the shadowing result for strong-concavity, along with the computation of the initial condition for the shadow in these subspaces. We summarize this result in the next theorem, which we prove formally in App. C.3. To enhance understanding, we illustrate the procedure of construction of a shadow in Fig. 2.

**Proposition 4.** *Let $f : \mathbb{R}^d \to \mathbb{R}$ be quadratic centered at $x^*$ with Hessian $H$ with no eigenvalues in the interval $(-\gamma, \mu)$, for some $\mu, \gamma > 0$. Assume the orbit $(y_k)_{k=0}^\infty$ of $\varphi_h^{GD}$ is s.t. (H1) holds up to iteration $K$. Let $\epsilon$ be the desired tracking accuracy; if $0 < h \le \min\left\{ \frac{\mu\epsilon}{L\ell}, \frac{\gamma\epsilon}{2L\ell}, \frac{1}{L} \right\}$, then $(y_k)_{k=0}^\infty$ is $\epsilon$-shadowed by an orbit $(x_k)_{k=0}^\infty$ of $\Psi_h^{GD}$ up to iteration $K$.*

**General saddles.** In App. C.4 we take inspiration from the literature on the approximation of stiff ODEs near stationary points [36, 5, 32] and use Banach fixed-point theorem to generalize the result above to perturbed quadratic saddles $f + \phi$, where $\phi$ is required to be $L_\phi$-smooth with $L_\phi \le \mathcal{O}(\min\{\gamma, \mu\})$. This condition is intuitive, since $\phi$ effectively counteracts the contraction/expansion.

**Theorem 4.** *Let $f : \mathbb{R}^d \to \mathbb{R}$ be a quadratic centered at $x^*$ with Hessian $H$ with no eigenvalues in the interval $(-\gamma, \mu)$, for some $\mu, \gamma > 0$. Let $g : \mathbb{R}^d \to \mathbb{R}$ be our objective function, of the form $g(x) = f(x) + \phi(x)$ with $\phi : \mathbb{R}^d \to \mathbb{R}$ a $L_\phi$-smooth perturbation such that $\nabla\phi(x^*) = 0$. Assume the orbit $(y_k)_{k=0}^\infty$ of $\varphi_h^{GD}$ on $g$ is s.t. (H1) (stated for g) holds, with gradients bounded by $\ell$ up to iteration $K$. Assume $0 < h \le \frac{1}{L}$ and let $\epsilon$ be the desired tracking accuracy, if also*

$$h \le \frac{\epsilon\left(\min\left\{\frac{\gamma}{2}, \mu\right\} - 4L_\phi\right)}{2\ell L},$$

*then $(y_k)_{k=0}^\infty$ is $\epsilon$-shadowed by a orbit $(x_k)_{k=0}^\infty$ of $\Psi_h^{GD}$ on $g$ up to iteration $K$.*

We note that, in the special case of strongly-convex quadratics, the theorem above recovers the shadowing condition of Cor. 1 up to a factor $1/2$ which is due to the different proof techniques.

**Gluing landscapes.** The last result can be combined with Thm. 3 to capture the dynamics of GD-ODE where directions of negative curvature are encountered during the early stage of training followed by a strongly-convex regions as we approach a local minimum (such as the one in Fig. 3). Note that, since under (H1) the objective is $C^2$, there will be a few iterations in the "transition phase" (non-convex to convex) where the curvature is very close to zero. These few iterations are not captured by Thm. 3 and Thm. 4; indeed, the error behaviour in Fig. 3 is pathological at $k \approx 10$. Nonetheless, as we showed for the convex case in Sec. 3.1, the approximation error during these iterations only grows as $\mathcal{O}(kh)$. In the numerical analysis literature, the procedure we just sketched was made precise in [11], who proved that a gluing argument is successful if the number of unstable directions on the ODE path is non-increasing.

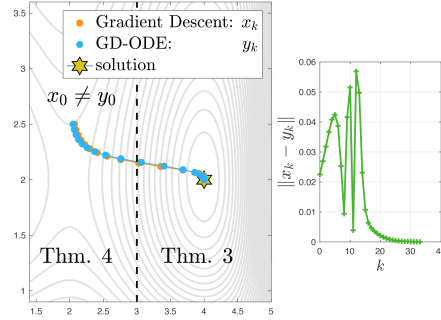

Figure 3: Dynamics on the Hosaki function, $h = 0.3$ and lightly perturbed initial condition in the unstable subspace. ODE numerical simulation with Runge-Kutta 4 method [21].

## 4 The Heavy-ball ODE

We now turn our attention to analyzing Heavy-ball whose continuous representation is $\ddot{q} + \alpha\dot{q} + \nabla f(q) = 0$, where $\alpha$ is a positive number called the *viscosity parameter*. Following [37], we introduce the velocity variable $p = \dot{q}$ and consider the dynamics of $y = (q, p)$ (i.e. in *phase space*).

$$\begin{cases} \dot{p} = -\alpha p - \nabla f(q) \\ \dot{q} = p \end{cases} \tag{HB-ODE}$$

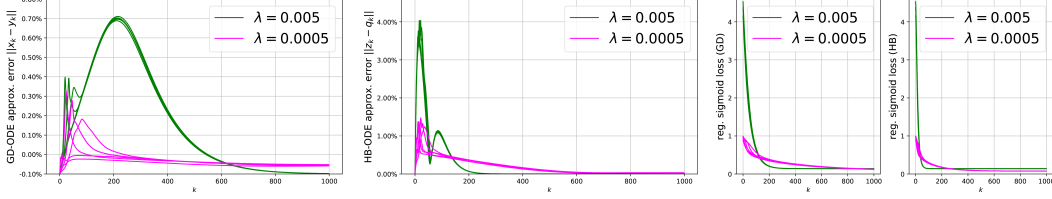

Figure 5: Shadowing results under the sigmoid loss in MNIST (2 digits). We show 5 runs for the ODE and for the algorithm, with same (random) initialization. ODEs are simulated with 4th-order RK: our implementation uses 4 back-propagations and an integrator-step of $0.1$. When trying higher precisions, results do not change. Shown are also the strictly decreasing (since we use full gradients) losses for each run the algorithms. The loss of the discretized ODEs are indistinguishable (because of shadowing) and are therefore not shown. We invite the reader to compare the results (in particular, for high $\lambda$) to the ones obtained in synthetic examples in Fig. 1 and 4.

Under (**H1**), we denote by $\varphi_{\alpha,h}^{\text{HB}} : \mathbb{R}^{2d} \to \mathbb{R}^{2d}$ the corresponding joint time-$h$ map and by $(y_k)_{k=0}^{\infty} = ((p_k, q_k))_{k=0}^{\infty}$ its orbit (i.e. the sampled HB-ODE trajectory). First, we show that semi-implicit[10] integration of Eq. (HB-ODE) yields HB.

Given a point $x_k = (v_k, z_k)$ in phase space, this integrator computes $(v_{k+1}, z_{k+1}) \cong \varphi_{\alpha,h}^{\text{HB}}(x_k)$ as

$$\begin{cases} v_{k+1} = v_k + h(-\alpha v_k - \nabla f(z_k)) \\ z_{k+1} = z_k + h v_{k+1} \end{cases} \tag{HB-PS}$$

Notice that $v_{k+1} = (z_{k+1} - z_k)/h$ and $z_{k+1} = z_k - (1 - \alpha h)(z_k - z_{k-1}) - h^2 \nabla f(z_k)$, which is exactly one iteration of HB, with $\beta = 1 - h\alpha$ and $\eta = h^2$. We therefore have established a numerical link between HB and HB-ODE, similar to the one presented in [47]. In the following, we use $\Psi_{\alpha,h}^{\text{HB}}$ to denote the one step map in phase space defined by HB-PS.

Similarly to Remark 1, by Prop. 2.2 in [8], (**H1**) implies that gradients are bounded by a constant $\ell$. Hence, we can get an analogue to Prop. 2 (see App. D.2 for the proof).

**Proposition 5.** *Assume (H1) and let $y_0 = (0, z_0)$. Then, $(y_k)_{k=0}^{\infty}$ is a $\delta$-pseudo-orbit of $\Psi_{\alpha,h}^{HB}$ with*

$$\delta = \ell(\alpha + 1 + L)h^2.$$

**Strong-convexity.** The next step, as done for GD, would be to consider strongly-convex landscapes and derive a formula for the shadowing radius (see Thm. 3). However — it is easy to realize that, in this setting, *HB is not uniformly contracting*. Indeed, it notoriously is *not a descent method*. Hence, it is unfortunately difficult to state an equivalent of Thm. 3 using similar arguments. We believe that the reason behind this difficulty lies at the very core of the *acceleration* phenomenon. Indeed, as noted by [20], the current known bounds for HB in the strongly-convex setting might be loose due to the tediousness of its analysis [46]— which is also reflected here. Hence, we leave this theoretical investigation (as well as the connection to acceleration and symplectic integration [47]) to future research, and show instead experimental results in Sec. 5 and Fig. 4.

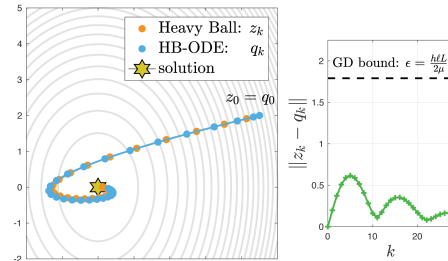

Figure 4: Orbit of the space variable in $\Psi_h^{\text{HB}}$, $\varphi_h^{\text{HB}}$ (sampled ODE solution) on a strongly-convex quadratic with $h = 0.2$ and $\alpha = 1$. Solution to HB-ODE was computed analytically.

**Quadratics.** In App. D.3 we show that, if $f$ is quadratic, then $\Psi_{\alpha,h}^{\text{HB}}$ is linear hyperbolic. Hence, as discussed in the introduction and thanks to Prop. 5, there exists a norm[11] under which we have shadowing, and we can recover a result analogous to Prop. 4 and to its perturbed variant (App. C.4). We show this empirically in Fig. 4 and compare with the GD formula for the shadowing radius.

# 5  Experiments on empirical risk minimization

We consider the problem of binary classification of digits 3 and 5 from the MNIST data-set [33]. We take $n = 10000$ training examples $\{(a_i, l_i)\}_{i=1}^n$, where $a_i \in \mathbb{R}^d$ is the i-th image (in $\mathbb{R}^{785}$ adding a bias) and $l_i \in \{-1, 1\}$ is the corresponding label. We use the regularized sigmoid loss (non-convex) $f(x) = \frac{\lambda}{2}\|x\|^2 + \frac{1}{n}\sum_{i=1}^n \phi(\langle a_i, x\rangle l_i)$, $\phi(t) = \frac{1}{1+e^t}$. Compared to the cross-entropy loss (convex), this choice of $f$ often leads to better generalization [45]. For 2 different choices of $\lambda$, using the *full gradient*, we simulate GD-ODE using fourth-order Runge-Kutta[21] (high-accuracy integration) and run GD with learning rate $h = 1$, which yields a steady decrease in the loss. We simulate HB-ODE and run HB under the same conditions, using $\alpha = 0.3$ (to induce a significant momentum). In Fig. 5, we show the behaviour of the approximation error, measured in percentage w.r.t. the discretized ODE trajectory, until convergence (with accuracy around $95\%$). We make a few comments on the results.

1. Heavy regularization (in green) increases the contractiveness of GD around the solution, yielding a small approximation error (it converges to zero) after a few iterations — exactly as in Fig. 1. For a small $\lambda$ (in magenta), the error between the trajectories is bounded but is slower to converge to zero, since local errors tend not to be corrected (cf. discussion for convex objectives in Sec. 3.1).

2. Locally, as we saw in Prop. 2, large gradients make the algorithm deviate significantly from the ODE. Since regularization increases the norm of the gradients experienced in early training, a larger $\lambda$ will cause the approximation error to grow rapidly at the first iterations (when gradients are large). Indeed, Cor. 1 predicts that the shadowing radius is proportional[12] to $\ell$.

3. Since HB has momentum, we notice that indeed it converges faster than GD [50]. As expected, (see point 2) this has a bad effect on the global shadowing radius, which is 5 times bigger. On the other hand, the error from HB-ODE is also much quicker to decay to zero when compared to GD.

Last in Fig. 6 we explore the effect of the shadowing radius $\epsilon$ on the learning rate $h$ and find a good match with the prediction of Cor. 1. Indeed, the experiment confirms that such relation is linear: $\epsilon = \mathcal{O}(h)$, with *no dependency on the number of iterations* (as opposed to the classical results discussed in the introduction).

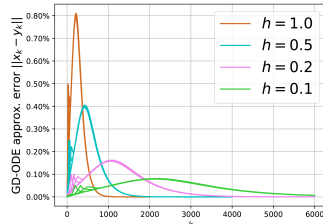

Figure 6: Approx. error under same setting of Fig. 5 for $\lambda = 0.005$. Experiment validates the *linear* dependency on $h$ in Cor. 1

All in all, we conclude from the experiments that the intuition developed from our analysis can potentially explain the behaviour of the GD-ODE and the HB-ODE approximation in simple machine learning problems.

# 6  Conclusion

In this work we used the theory of shadowing to motivate the success of continuous-time models in optimization. In particular, we showed that, if the cost $f$ is strongly-convex or hyperbolic, then any GD-ODE trajectory is shadowed by a trajectory of GD, with a slightly perturbed initial condition. Weaker but similar results hold for HB. To the best of our knowledge, this work is the first to provide this type of quantitative link between ODEs and algorithms in optimization. Moreover, our work leaves open a lot of directions for future research, including the derivation of a formula for the shadowing radius of Heavy-ball (which will likely give insights on acceleration), the extension to other algorithms (e.g. Newton's method) and to stochastic settings. Actually, a partial answer to the last question was provided in the last months for the strongly-convex case in [17]. In this work, the authors use *backward error analysis* to study how close SGD is to its approximation using a high order (involving the Hessian) stochastic modified equation. It would be interesting to derive a similar result for a stochastic variant of GD-ODE, such as the one studied in [40].

**Acknowledgements.**  We are grateful for the enlightening discussions on numerical methods and shadowing with Prof. Lubich (in Tübingen), Prof. Hofmann (in Zürich) and Prof. Benettin (in Padua). Also, we would like to thank Gary Bécigneul for his help in completing the proof of hyperbolicity of Heavy-ball and Foivos Alimisis for pointing out a mistake in the initial draft.

## Footnotes

*Correspondence to `orvietoa@ethz.ch`

[2]The error between the numerical approximation and the actual trajectory with the same initial conditions is, for a $p$-th order method, $e^{Ct}h^p$ at time t with $C \gg 0$.

[3]Formally defined in Sec. 2.

[4]A $C^1$ map with well-defined and $C^1$ inverse.

[5]This case is restrictive, yet it includes the important cases of the dynamics of GD and HB when $f$ is a quadratic (which is the case in, for instance, least squares regression).

[6]For a precise description of this similarity, we invite the reader to read on *topological conjugacy* in [2].

[7]$f(x) \to \infty$ as $\|x\| \to \infty$

[8]This in line with the requirement of hyperbolicity in the shadowing theory: a convex function might have zero curvature, hence the corresponding gradient system is going to have an eigenvalue on the unit circle.

[9]Indeed, Thm. 3 applied backward in time from $y_K$ ensures that $\|(\Psi_h^{\text{GD}})^{-K}(y_K) - x_0\| \leq \epsilon$.

[10]Note that this integrator, when applied to a Hamiltonian system, is *symplectic* (see e.g. definition in [21]).

[11]For GD, this was the Euclidean norm. For HB the norm we have to pick is different, since (differently from GD) $\varphi_{\alpha,h}^{\text{HB}}(x) = Ax$ with $A$ non-symmetric. The interested reader can find more information in App. 4 of [24].

[12]Alternatively, looking at the formula $\epsilon = \frac{hL\ell}{2\mu}$ in Cor. 1 and noting $\ell \leq L\|x_0 - x^*\|$, we get $\epsilon = \mathcal{O}(L^2/\mu)$. Hence, regularization, which increases $L$ and decreases $\mu$ by the same amount, actually increases $\epsilon$.

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
