[Supplementary Material]

# Appendix

## A    Notation and problem definition

We work in $\mathbb{R}^d$ with the Euclidean norm $\|\cdot\|$. If $A$ is a matrix, $\|A\|$ denotes its supremum norm: $\|A\| = \sup_{\|y\|=1} \|Ay\|$. We denote by $C^r(\mathbb{R}^n, \mathbb{R}^m)$ the family of $r$-times continuously differentiable functions from $\mathbb{R}^n$ to $\mathbb{R}^m$. We denote by $Dg \in \mathbb{R}^{m \times n}$ the Jacobian (matrix of partial derivatives) of $g: \mathbb{R}^n \to \mathbb{R}^m$; if $m = 1$ then we denote its gradient by $\nabla g$. If the dimensions are clear, we write $C^r$.

We seek to minimize $f: \mathbb{R}^d \to \mathbb{R}$. We list below some assumptions we will refer to.

**(H1)**  $f \in C^2(\mathbb{R}^d, \mathbb{R})$ is coercive[13], bounded from below and $L$-smooth ($\forall a \in \mathbb{R}^d, \|\nabla^2 f(a)\| \le L$).

**(H2)**  $f$ is $\mu$-strongly-convex: for all $a \in \mathbb{R}^d$, $\|\nabla^2 f(a)\| \ge \mu$.

## B    Additional results on shadowing

We state here some useful details on shadowing for the interested reader. Also, we propose a simple proof of the expansion map shadowing theorem based on [39].

### B.1    Shadowing near hyperbolic sets

We provide the definitions and results needed to state precisely the shadowing theorem [1]. Our discussion is based on [31]. Let $\Psi: \mathbb{R}^n \to \mathbb{R}^n$ be a diffeomorphism.

**Definition 4.** *We say $x \in \mathbb{R}^n$ is an **hyperbolic point** for $\Psi$ if there exist a splitting $\mathbb{R}^n = E_s(x) \oplus E_u(x)$ in linear subspaces such that*

Figure 7: (From Wikipedia) The cat map stretches the unit square and how its pieces are rearranged. The cat map is Anosov. Shown are the directions of the *global* splitting $\mathbb{R}^n = E_s \oplus E_u$.

$$\|D(\Psi^k(x))\xi\| \le c\lambda^k \|\xi\| \quad \text{for all } \xi \in E_s(x) \text{ and for all } k \in \mathbb{N},$$
$$\|D(\Psi^{-k}(x))\xi\| \le c\lambda^k \|\xi\| \quad \text{for all } \xi \in E_u(x) \text{ and for all } k \in \mathbb{N},$$

*where $\lambda$ and $c$ can be taken to depend only on $x$, not on $\xi \in E_u(x)$.*

**Remark 2.** *It can be showed that, if $x$ is hyperbolic for $\Psi$, then also $\Psi(x)$ and $\Psi^{-1}(x)$ are hyperbolic points. Moreover, using the chain rule, we have*

$$E_s(\Psi(x)) = D\Psi(x)E_s(x) \text{ and } E_u(\Psi(x)) = D\Psi(x)E_u(x).$$

**Definition 5.** *A compact set $\Lambda \subset \mathbb{R}^n$ is said to be an **hyperbolic set** for $\Psi$ if it is invariant for $\Psi$ (i.e. $\Psi(\Lambda) = \Lambda$), each $x \in \Lambda$ is hyperbolic and $c, \lambda$ (see Def. 4) can be taken to be independent of $x$. If the entire space is hyperbolic, then $\Psi$ is called an **Anosov diffeomorphism** (from [1]).*

We now state the main result in the literature, originally proved in [6], which essentially tells us that pseudo-orbits sufficiently near an hyperbolic set are shadowed.

> **Theorem B.1** (Shadowing theorem). *Let $\Lambda$ be an hyperbolic set for $\Psi$, and let $\epsilon > 0$. Then there exists $\delta > 0$ such that, for every $\delta$-pseudo-orbit $(y_k)_{k=0}^{\infty}$ with $\|y_k - \Lambda\| \le \delta$ for all $k$, there is $x_0$ such that*
> $$\|y_k - \Psi^k(x_0)\| \le \epsilon \text{ for all } k \in \mathbb{N}.$$
> *Furthermore, if $\epsilon$ is small enough, $x_0$ is unique.*

**Example.**    The most famous example of an Anosov diffeomorphism is Arnold's cat map [7] on the torus $\mathbb{T}^2$, which can be thought of as the quotient space $\mathbb{R}^2/\mathbb{Z}^2$. Shown in Fig. 7, details in [7].

## B.2 Expansion map shadowing theorem

First, we remind to the reader an important result in analysis.

> **Theorem B.2** (Banach fixed-point theorem). *Let $(Z, d)$ be a non-empty complete metric space with a contraction mapping $T : Z \to Z$. Then $T$ admits a unique fixed-point $z^* \in Z$ (i.e. $T(z^*) = z^*$). Furthermore, $z^*$ can be found as follows: start with an arbitrary element $z_0$ in $Z$ and iteratively apply $T$; $z^*$ is the limit of this sequence.*

We recall that $\Psi$ is said to be **uniformly expanding** if there exists $\rho > 1$ (*expansion factor*) such that for all $x_1, x_2 \in \mathbb{R}^n$, $\|\Psi(x_1) - \Psi(x_2)\| \geq \rho \|x_1 - x_2\|$. The next result is adapted from Prop. 1 in [39].

> **Theorem B.3** (Expansion map shadowing theorem). *If $\Psi$ is uniformly expanding, then for every $\epsilon > 0$ there exists $\delta > 0$ such that every $\delta-$pseudo-orbit $(y_k)_{k=0}^{\infty}$ of $\Psi$ is $\epsilon-$shadowed by the orbit $(x_k)_{k=0}^{\infty}$ of $\Psi$ starting at $x_0 = \lim_{k \to \infty} \Psi^{-k}(y_k)$, that is $x_k := \Psi^k(x_0)$. Moreover,*
>
> $$\delta \leq \left(1 - \frac{1}{\rho}\right)\epsilon. \tag{1}$$

*Proof.* Fix $\epsilon > 0$ and define $\delta = (1 - 1/\rho)\epsilon$. Let $(y_k)_{k=0}^{\infty}$ be a $\delta$-pseudo-orbit of $\Psi$ and extend it to negative iterations: $y_{-k} := \Psi^{-k}(y_0)$. We call the resulting sequence $y = (y_k)_{k \in \mathbb{Z}}$. We consider the set $Z$ of sequences which are pointwise close to $y$:

$$Z = \{z : z = (z_k)_{k \in \mathbb{Z}}, \|z_k - y_k\| \leq \epsilon \text{ for all } k \in \mathbb{Z}\}.$$

We endow $Z$ with the supremum metric $d$, defined as follows:

$$d(z, w) = \sup_{k \in \mathbb{Z}} \|z_k - w_k\|.$$

It is easy to show that $(Z, d)$ is complete. Next, we define an operator $T$ on sequences in $Z$, such that

$$[T(z)]_k = \Psi^{-1}(z_{k+1}) \text{ for all } k \in \mathbb{Z}.$$

Notice that, if $T(x) = x$, then $(x_k)_{k=0}^{\infty}$ is an orbit of $\Psi$. If, in addition $x \in Z$, then by definition we have that $(x_k)_{k=0}^{\infty}$ shadows $(y_k)_{k=0}^{\infty}$. We clearly want to apply Thm. B.2, and we need to verify that

1. $T(Z) \subseteq Z$. This follows from the fact that $\Psi^{-1}$ is a contraction with contraction factor $1/\rho$, similarly to the contraction map shadowing theorem (see Thm. 2). Let $z \in Z$, for all $k \in \mathbb{Z}$

$$
\begin{aligned}
\|\Psi^{-1}(z_{k+1}) - y_k\| &\leq \|\Psi^{-1}(z_{k+1}) - \Psi^{-1}(y_{k+1})\| + \|\Psi^{-1}(y_{k+1}) - y_k\| \\
&\overset{\delta\text{-pseudo-orbit}}{\leq} \|\Psi^{-1}(z_{k+1}) - \Psi^{-1}(y_{k+1})\| + \delta \\
&\overset{\text{contraction}}{\leq} \frac{1}{\rho}\|(z_{k+1}) - y_{k+1}\| + \delta \\
&\overset{z \in Z}{\leq} \frac{1}{\rho}\epsilon + \delta = \epsilon.
\end{aligned}
$$

2. $T$ is itself a contraction in $(Z, d)$, since

$$
\begin{aligned}
d(T(z), T(w)) &= \sup_{k \in \mathbb{Z}} \|\Psi^{-1}(z_{k+1}) - \Psi^{-1}(w_{k+1})\| \\
&\leq \frac{1}{\rho} \sup_{k \in \mathbb{Z}} \|z_{k+1} - w_{k+1}\| \\
&\leq \frac{1}{\rho} d(z, w).
\end{aligned}
$$

The statement of the expansion map shadowing theorem follows then directly from the Banach fixed-point theorem applied to $T$, which is a contraction on $(Z, d)$. ∎

# C  Shadowing in optimization

We present here the proofs of some results we claim in the main paper.

## C.1  Non-expanding maps (i.e. the convex setting)

> **Proposition C.1.** *If $\Psi$ is uniformly non-expanding, then for $\delta-$pseudo-orbit $(y_k)_{k=0}^{\infty}$ of $\Psi$ is such that the orbit $(x_k)_{k=0}^{\infty}$ of $\Psi$ starting at $x_0 = y_0$, satisfies $\|x_k - y_k\| \leq \delta k$ for all $k$.*

*Proof.* The proposition is trivially true at $k = 0$; next, we assume the proposition holds at $k \in \mathbb{N}$ and we show validity for $k + 1$. We have

$$
\begin{aligned}
\|x_{k+1} - y_{k+1}\| &\overset{\text{subadditivity}}{\leq} \|\Psi(x_k) - \Psi(y_k)\| + \|\Psi(y_k) - y_{k+1}\| \\
&\overset{\delta\text{-pseudo-orbit}}{\leq} \|\Psi(x_k) - \Psi(y_k)\| + \delta \\
&\overset{\text{non-expansion}}{\leq} \|x_k - y_k\| + \delta \\
&\overset{\text{induction}}{\leq} k\delta + \delta = (k+1)\delta.
\end{aligned}
$$

∎

The GD map on a convex function is non-expanding, hence this result bounds the GD-ODE approximation, which grows slowly as a function of the number of iterations.

## C.2  Perturbed contracting maps (i.e. the stochastic gradients setting)

> **Proposition C.2.** *Assume $\Psi$ is uniformly contracting with contraction factor $\rho$. Let $\tilde{\Psi}$ be the perturbed dynamical system, that is $\tilde{\Psi}(x) = \Psi(x) + \zeta$ where $\|\zeta\| \leq D$. Let $(y_k)_{k=0}^{\infty}$ be a $\delta-$pseudo-orbit of $\Psi$ and we denote by $(\tilde{x}_k)_{k=0}^{\infty}$ the orbit of of $\tilde{\Psi}$ starting at $\tilde{x}_0 = y_0$, that is $\tilde{x}_k := \tilde{\Psi}^k(\tilde{x}_0)$. We have $\epsilon$-shadowing under*
> $$\delta \leq (1 - \rho)\epsilon - D.$$

*Proof.* Again as in Thm. 2, we proceed by induction: the proposition is trivially true at $k = 0$, since $\|\tilde{x}_0 - y_0\| \leq \epsilon$; next, we assume the proposition holds at $k \in \mathbb{N}$ and we show validity for $k + 1$. We have

$$
\begin{aligned}
\|\tilde{x}_{k+1} - y_{k+1}\| &\overset{\text{subadditivity}}{\leq} \|\Psi(\tilde{x}_k) + \zeta - \Psi(y_k)\| + \|\Psi(y_k) - y_{k+1}\| \\
&\overset{\delta\text{-pseudo-orbit}}{\leq} \|\Psi(\tilde{x}_k) - \Psi(y_k)\| + D + \delta \\
&\overset{\text{contraction}}{\leq} \rho\|\tilde{x}_k - y_k\| + D + \delta \\
&\overset{\text{induction}}{\leq} \rho\epsilon + D + \delta.
\end{aligned}
$$

Since $\delta \leq (1 - \rho)\epsilon - D$, $\rho\epsilon + \delta + D = \epsilon$. ∎

In the setting of shadowing GD, $\rho = 1 - \mu h$ (Prop. 3), $\delta = \frac{\ell L h^2}{2}$, and $D = h\|\nabla f(x) - \tilde{\nabla} f(x)\| \leq Rh$ which gives the consistency equation

$$
\frac{\ell L h^2}{2} \leq (1 - \rho)\epsilon - D \leq \mu h \epsilon - Rh \implies \boxed{h \leq \frac{2(\epsilon\mu - R)}{\ell L}}.
$$

## C.3 Hyperbolic maps (i.e. the quadratic saddle setting)

We prove the result for GD. This can be generalized to HB. A more powerful proof technique — which can tackle the effect of perturbations — can be found in App. C.4.

> **Proposition C.3** (restated Prop. 4). *Let $f$ be quadratic with Hessian $H$ which has no eigenvalues in the interval $(-\gamma, \mu)$, for some $\mu, \gamma > 0$. Assume the orbit $(y_k)_{k=0}^{\infty}$ of $\varphi_h^{GD}$ is such that (**H1**) holds up to iteration $K$. Let $\epsilon$ be the desired tracking accuracy; if $0 < h \leq \min\left\{\frac{\mu\epsilon}{L\ell}, \frac{\gamma\epsilon}{2L\ell}, \frac{1}{L}\right\}$, then $(y_k)_{k=0}^{\infty}$ is $\epsilon$-shadowed by a orbit $(x_k)_{k=0}^{\infty}$ of $\Psi_h^{GD}$ up to iteration $K$.*

*Proof.* From Prop. 2 and thanks to the hypothesis, we know $(y_k)_{k=0}^{K}$ is a partial $\delta$-pseudo-orbit of $\Psi_h^{GD}$ with $\delta = \frac{\ell L}{2}h^2$. By the triangle inequality, this property is preserved when projecting both sequences on a subspace. We project the sequences (the orbit and the pseudo-orbit) onto the stable and unstable subspaces ($E_s$ and $E_u$) of $H$. Since these are invariant (they are eigenspaces), shadowing in each space implies shadowing in the whole $\mathbb{R}^d$ [39]. On the stable subspace we require, by the same argument of Thm. 3, $\delta \leq \mu h \epsilon$ which gives the condition $h \leq \frac{2\epsilon\mu}{\ell L}$. On the unstable space, reversing the arrow of time (see discussion in the main paper) and by Thm. B.3, we require $\delta \leq \left(1 - \frac{1}{1+\gamma h}\right)\epsilon = \frac{\gamma h}{1+\gamma h}\epsilon$. Since $\gamma \leq L$ and $h \leq \frac{1}{L}$, we have $\gamma h \leq 1$; which implies[14] $\frac{\gamma h}{2}\epsilon \leq \frac{\gamma h}{1+\gamma h}\epsilon$. An easier but stronger sufficient condition is therefore $\delta \leq \frac{\gamma h}{2}\epsilon$. Therefore, shadowing in the unstable subspace requires $\frac{\ell L}{2}h^2 \leq \frac{\gamma h}{2}\epsilon \implies h \leq \frac{\gamma\epsilon}{\ell L}$. All in all, since we have to consider both manifold together, by subadditivity of the norm we actually need to require a radius of $\epsilon/2$. ∎

## C.4 General saddle points

We prove the result for GD. This can be generalized to HB. It's interesting to compare the result below with the one in App. C.2: *a perturbation on the landscape is essentially a perturbation on the gradient.*

> **Theorem C.1.** *Let $f : \mathbb{R}^d \to \mathbb{R}$ be a quadratic centered at $x^*$ with Hessian $H$ with no eigenvalues in the interval $(-\gamma, \mu)$, for some $\mu, \gamma > 0$. Let $g : \mathbb{R}^d \to \mathbb{R}$ be our objective function, of the form $g(x) = f(x) + \phi(x)$ with $\phi : \mathbb{R}^d \to \mathbb{R}$ a $L_\phi$−smooth perturbation such that $\nabla\phi(x^*) = 0$. Assume the orbit $(y_k)_{k=0}^{\infty}$ of $\varphi_h^{GD}$ on $g$ is s.t. (**H1**) (stated for $g$) holds, with gradients bounded by $\ell$ up to iteration $K$. Assume $0 < h \leq \frac{1}{L}$ and let $\epsilon$ be the desired tracking accuracy, if also*
> $$h \leq \frac{\epsilon\left(\min\left\{\frac{\gamma}{2}, \mu\right\} - 4L_\phi\right)}{2\ell L},$$
> *then $(y_k)_{k=0}^{\infty}$ is $\epsilon$-shadowed by a orbit $(x_k)_{k=0}^{\infty}$ of $\Psi_h^{GD}$ on $g$ up to iteration $K$.*

We follow the proof technique presented in [36] for integration of stiff equations (this paper generalizes an important result of Beyn [5] for approximation of phase portraits).

*Proof.* We first introduce some notation and a change of basis, then proceed using Thm. B.2.

Preparation: let $f(x) = \langle x - x^*, H(x - x^*) \rangle$, with $H$ with no eigenvalues in the interval $(-\gamma, \mu)$, perturbed by a nonlinearity $\phi(x)$. Gradient Descent on such function is the dynamical system:

$$\Psi_h^{GD}(x) = x - h\nabla f(x) + h\nabla\phi(x - x^*).$$

Without loss of generality, we assume $x^* = 0$ and end up with the map

$$\Psi_h^{GD}(x) = R(hH)x + h\nabla\phi(x),$$

where $R(hH) := I - hH$. Next, let the rows of $V$ denote a new basis which block diagonalizes $H$ into the positive and negative parts of the spectrum; we denote these blocks by $H^+$ and $H^-$, respectively. This change of basis also block-diagonalizes $R(hH)$:

$$VR(hH)V^T = \begin{pmatrix} R(hH^+) & 0 \\ 0 & R(hH^-) \end{pmatrix}.$$

To work in this basis, we follow [36] and define

$$x =: V\begin{pmatrix} x^+ \\ x^- \end{pmatrix}, \quad \nabla\phi =: V\begin{pmatrix} \nabla\phi^+ \\ \nabla\phi^- \end{pmatrix}, \quad \Psi_h^{\text{GD}} =: V\begin{pmatrix} \Psi_h^{\text{GD}+} \\ \Psi_h^{\text{GD}-} \end{pmatrix}.$$

Hence,

$$V\begin{pmatrix} \Psi_h^{\text{GD}+}(x) \\ \Psi_h^{\text{GD}-}(x) \end{pmatrix} = R(hH)V\begin{pmatrix} x^+ \\ x^- \end{pmatrix} + V\begin{pmatrix} \nabla\phi^+(x) \\ \nabla\phi^-(x) \end{pmatrix}.$$

Also, since $V$ is orthonormal,

$$\begin{pmatrix} \Psi_h^{\text{GD}+}(x) \\ \Psi_h^{\text{GD}-}(x) \end{pmatrix} = \begin{pmatrix} R(hH^+) & 0 \\ 0 & R(hH^-) \end{pmatrix}\begin{pmatrix} x^+ \\ x^- \end{pmatrix} + h\begin{pmatrix} \nabla\phi^+(x) \\ \nabla\phi^-(x) \end{pmatrix}. \tag{2}$$

Note that also time-$h$ map of GD-ODE (see Section 3) can be written as a perturbation (quantified by a function $\zeta : \mathbb{R}^d \to \mathbb{R}^d$) of the system above:

$$\begin{pmatrix} \varphi_h^{\text{GD}+}(x) \\ \varphi_h^{\text{GD}-}(x) \end{pmatrix} = \begin{pmatrix} R(hH^+) & 0 \\ 0 & R(hH^-) \end{pmatrix}\begin{pmatrix} x^+ \\ x^- \end{pmatrix} + h\begin{pmatrix} \nabla\phi^+(x) \\ \nabla\phi^-(x) \end{pmatrix} + \begin{pmatrix} \zeta^+(x) \\ \zeta^-(x) \end{pmatrix}, \tag{3}$$

where $\zeta =: V\begin{pmatrix} \zeta^+ \\ \zeta^- \end{pmatrix}$.

Let $(y_k)_{k=0}^K$ be the partial orbit of $\varphi_h^{\text{GD}}$ which we want to shadow with an orbit of $\Psi_h^{\text{GD}}$. We work in the basis $V$ defined above, i.e. we consider $((y_k^+, y_k^-))_{k=0}^K$ where $y_{k+1}^+ = \varphi_h^{\text{GD}+}(y_k)$ and $y_{k+1}^- = \varphi_h^{\text{GD}-}(y_k)$.

Before proceeding with the proof, we note the following:

- Starting from Eq. (3), we have that $((y_k^+, y_k^-))_{k=0}^K$ can be written recursively using the discrete variations of constants formula (cf. page 6 in [36]), forwards

$$y_k^- = R(hH^-)^k y_0^- + h\sum_{j=0}^{k-1} R(hH^-)^{k-j-1}\nabla\phi^-(y_j) + h\sum_{j=0}^{k-1} R(hH^-)^{k-j-1}\zeta^-(y_j) \tag{4}$$

  and backwards

$$y_k^+ = R(hH^+)^{k-K}y_K^+ - h\sum_{j=k}^{K-1} R(hH^+)^{k-j-1}\nabla\phi^+(y_j) - h\sum_{j=k}^{K-1} R(hH^+)^{k-j-1}\zeta^+(y_j). \tag{5}$$

  Any orbit of Gradient Descent also satisfies the equations above, with $\zeta^+ = 0$. We are going to use this fact later to build a contracting operator on the space of sequences near $(y_k)_{k=0}^K$.

- under (H1), both $\|\zeta^+(x)\|$ and $\|\zeta^-(x)\|$ are bounded by the pseudo-orbit error in the original basis: thanks to Prop. 2, for all $x \in \mathbb{R}^d$,

$$\|\zeta^\pm(x)\| \leq \left\|\begin{pmatrix} \zeta^+(x) \\ \zeta^-(x) \end{pmatrix}\right\| \overset{V \text{ orthonormal}}{=} \left\|V\begin{pmatrix} \zeta^+(x) \\ \zeta^-(x) \end{pmatrix}\right\| = \|\zeta(x)\| \leq \frac{\ell L}{2}h^2.$$

- Since $\phi$ is $L_\phi$-smooth, also $\nabla\phi^+$ and $\nabla\phi^-$ are Lipschitz with constant $L_\phi$: for all $x, y \in \mathbb{R}^d$, in the same way as above,

$$\begin{aligned}
\|\nabla\phi^\pm(x) - \nabla\phi^\pm(x)\| &\leq \left\|\begin{pmatrix} \nabla\phi^+(x) - \nabla\phi^+(y) \\ \nabla\phi^-(x) - \nabla\phi^-(y) \end{pmatrix}\right\| \\
&= \left\|V\begin{pmatrix} \nabla\phi^+(x) - \nabla\phi^+(y) \\ \nabla\phi^-(x) - \nabla\phi^-(y) \end{pmatrix}\right\| \\
&= \|\nabla\phi(x) - \nabla\phi(x)\| \\
&\leq L_\phi\|x - y\|.
\end{aligned}$$

Application of the fixed point theorem: We build a set of sequences close to $(y_k)_{k=0}^K$:

$$Z = \left\{ z : z = (z_k)_{k=0}^K \text{ s.t. } \|z_k - y_k\| \leq \epsilon \text{ for all } k = 0, \cdots, K \quad \text{and} \quad z_K^+ = x_K^+, z_0^- = x_0^- \right\},$$

where

- the superscripts indicate projection onto the stable ($-$) and unstable ($+$) subspaces of $H$, as clear from the paragraph above;

- $x_K^+$ and $x_0^-$ are the final and initial condition on the stable and unstable subspaces. We allow some freedom[15] in the initial condition: $\|x_K^+ - y_K^+\| + \|x_0^- - y_0^-\| \leq \epsilon/2$.

We want to show that there exists an orbit of gradient descent in this set. As a first step, we endow $Z$ with the supremum metric $d$, defined as follows:

$$d(z, w) = \sup_{k \in \{0, \cdots, K\}} \|z_k - w_k\|.$$

Clearly, $(Z, d)$ is complete. We now define an operator $T$ which takes any sequence in $Z$ and *brings it closer to an orbit of gradient descent*. This operator is inspired by Thm. B.3, acts on the representation in the above defined basis $V$ and coincides with the propagation of gradient steps forwards and backwards (i.e. following Eq. (4) and Eq. (5) with $\zeta = 0$), in the stable and unstable subspaces of $R(hH)$, respectively:

$$[Tz]_k = V \begin{pmatrix} R(hH^+)^{k-K} z_K^+ - h \sum_{j=k}^{K-1} R(hH^+)^{k-j-1} \nabla \phi^+(z_j) \\ R(hH^-)^k z_0^- + h \sum_{j=0}^{k-1} R(hH^-)^{k-j-1} \nabla \phi^-(z_j) \end{pmatrix},$$

where $[Tz]_k$ is the $k$-th element of the sequence $Tz$. Clearly, if $x = (x_k)_{k=0}^K$ is an orbit of Gradient Descent, then it is left fixed from $T$, i.e. $Tx = x$. Therefore, if we prove that $T$ is invariant ($Tz \in Z$ for all $z \in Z$) and is a contraction (for all $z, w \in Z$, $d(Tz, Tw) \leq d(z, w)$), then by Banach's fixed point theorem (Thm. B.2) there exist a gradient descent orbit in $Z$ (i.e. close to $y$).

$\rightarrow$ We start by verifying the contraction property. Let $z, w \in Z$,

$$\|[Tz]_k - [Tw]_k\| = \|V^T([Tz]_k - [Tw]_k)\|$$

$$\leq \left\| R(hH^+)^{k-K}(z_K^+ - w_K^+) - h \sum_{j=k}^{K-1} R(hH^+)^{k-j-1}(\nabla \phi^+(z_j) - \nabla \phi^+(w_j)) \right\|$$

$$+ \left\| R(hH^-)^k(z_0^- - w_0^-) + h \sum_{j=0}^{k-1} R(hH^-)^{k-j-1}(\nabla \phi^-(z_j) - \nabla \phi^-(w_j)) \right\|$$

$$= \left\| h \sum_{j=k}^{K-1} R(hH^+)^{k-j-1}(\nabla \phi^+(z_j) - \nabla \phi^+(w_j)) \right\|$$

$$+ \left\| h \sum_{j=0}^{k-1} R(hH^-)^{k-j-1}(\nabla \phi^-(z_j) - \nabla \phi^-(w_j)) \right\|$$

$$\leq h \sum_{j=k}^{K-1} \left\| R(hH^+)^{k-j-1} \right\| \left\| \nabla \phi^+(z_j) - \nabla \phi^+(w_j) \right\|$$

$$+ h \sum_{j=0}^{k-1} \left\| R(hH^-)^{k-j-1} \right\| \left\| \nabla \phi^-(z_j) - \nabla \phi^-(w_j) \right\|$$

$$\leq h L_\phi \left( \sum_{j=k}^{K-1} \left\| R(hH^+)^{k-j-1} \right\| + \sum_{j=0}^{k-1} \left\| R(hH^-)^{k-j-1} \right\| \right) d(z, w),$$

where we used the triangle inequality (multiple times), the fact that $z_K^+ = w_K^+$ and $z_0^- = w_0^-$ and the properties of $\phi$. Next, since the operator norm is submultiplicative, we have

$$\|[Tz]_k - [Tw]_k\| \leq hL_\phi \left( \sum_{j=k}^{K-1} \|R(hH^+)^{-1}\|^{j+1-k} + \sum_{j=0}^{k-1} \|R(hH^-)\|^{k-j-1} \right) d(z,w).$$

Last, we need to bound $\|R(hH^+)\|$ and $\|R(hH^-)\|$.

- $\|R(hH^-)\|$ is the biggest (in absolute value) eigenvalue of $R(hH^-) = I - hH^+$ (since this matrix is symmetric[16]) which is bounded by $1 - \mu h$ under our assumption $h \leq 1/L$.

- The eigenvalues of $R(hH^+)^{-1}$ are the inverse of the eigenvalues of $R(hH^+)$. As for the stable part, since $R(hH^+)$ is symmetric and $h \leq 1/L$, $\|R(hH^+)^{-1}\| = \frac{1}{1+\gamma h}$. However, we will use the more convenient bound $\|R(hH^+)^{-1}\| \leq 1 - \frac{\gamma}{2}h$. The inequality comes from the fact that, for $x \leq 1$, $\frac{1}{1+x} \leq 1 - \frac{x}{2}$. We can apply this since $h \leq 1/L$, so $h \leq 1/\gamma$.

We define the *skewness* parameter $\rho$ to bound both operator norms

$$\rho := \max\left\{1 - \mu h, 1 - \frac{\gamma}{2}h\right\} = 1 - \min\left\{\mu, \frac{\gamma}{2}\right\}h.$$

Going back to the proof of contraction, we therefore have

$$\begin{aligned}
\|[Tz]_k - [Tw]_k\| &\leq hL_\phi \left( \sum_{j=k}^{K-1} \rho^{j+1-k} + \sum_{j=0}^{k-1} \rho^{k-j-1} \right) d(z,w). \\
&= hL_\phi \left( \sum_{m=1}^{K-k} \rho^m + \sum_{m=0}^{k-1} \rho^m \right) d(z,w). \\
&\leq 2hL_\phi \left( \sum_{j=0}^{\infty} \rho^j \right) d(z,w). \\
&= \frac{2hL_\phi}{1-\rho} d(z,w).
\end{aligned}$$

To have a contraction, we need $\frac{2hL_\phi}{1-\rho} < 1$, which is satisfied if $\rho < 1 - 2L_\phi h$. A quick comparison with the definition of $\rho$ leads to the condition

$$2L_\phi < \min\left\{\mu, \frac{\gamma}{2}\right\}. \tag{6}$$

This condition makes sense, indeed *the curvature of f has to be strong enough to fight the perturbation $\phi$, both in the stable and unstable subspaces.*

$\rightarrow$ Finally, we need to check invariance of $T$. Let $z \in Z$, we want to show that $T(z) \in Z$. We have

$$\|[Tz]_k - y_k\| \leq h \left\| \sum_{j=k}^{K-1} R(hH^+)^{k-j-1}(\nabla\phi^+(z_j) - \nabla\phi^+(y_j)) \right\|$$

$$+ h \left\| \sum_{j=0}^{k-1} R(hH^-)^{k-j-1}(\nabla\phi^-(z_j) - \nabla\phi^-(y_j)) \right\|$$

$$+ \left\| R(hH^-)^k(z_0^- - y_0^-) \right\| + \left\| R(hH^+)^{k-K}(z_K^+ - y_K^+) \right\|$$

$$+ \left\| \sum_{j=0}^{k-1} R(hH^-)^{k-j-1}\zeta^-(y_j) \right\| + \left\| \sum_{j=k}^{K-1} R(hH^+)^{k-j-1}\zeta^+(y_j) \right\|,$$

where this inequality follows the exact steps as for the contraction proof, with the only difference that we allow $z_0^- \neq y_0^-$ and $z_0^+ \neq y_0^+$ (so we have two extra error terms) and that $y$ evolves with an additional error $\zeta$, which leads to two extra terms similar to the ones involving $\nabla\phi$ (which is the error from the quadratic approximation). Using the triangle inequality, the bounds on the spectral norms and the computation for the first two terms (which are equal for the contraction proof),

$$\|[Tz]_k - y_k\| \leq \frac{2h}{1-\rho}\left(L_\phi d(z,y) + \frac{\ell L}{2}h\right) + \|z_K^+ - y_K^+\| + \|z_0^- - y_0^-\|.$$

By construction, we have $\|z_k - y_k\| \leq \epsilon$ for all $k = 0, \cdots, K$ and $\|z_K^+ - y_K^+\| + \|z_0^- - y_0^-\| \leq \epsilon/2$. Hence, a sufficient condition for having $\|[Tz]_k - y_k\| \leq \epsilon$ for all $k = 0, \cdots, K$ (which implies invariance), is

$$\frac{2h}{1-\rho}\left(L_\phi\epsilon + \frac{\ell L}{2}h\right) + \frac{\epsilon}{2} \leq \epsilon$$

$$\Longleftrightarrow h\left(L_\phi\epsilon + \frac{\ell L}{2}h\right) \leq \frac{\epsilon}{4}(1-\rho)$$

$$\Longleftrightarrow h\left(L_\phi\epsilon + \frac{\ell L}{2}h\right) \leq \frac{\epsilon}{4}\min\left\{\mu, \frac{\gamma}{2}\right\}h$$

$$\Longleftrightarrow 4L_\phi\epsilon + 2\ell L h \leq \epsilon\min\left\{\mu, \frac{\gamma}{2}\right\}$$

$$\Longleftrightarrow h \leq \frac{\min\left\{\mu, \frac{\gamma}{2}\right\} - 4L\phi}{2\ell L}.$$

Note that if this condition is satisfied for a positive $h$, then we automatically satisfy the condition for a contraction, i.e. Eq. (6).

Putting it all together, we found that if $0 < h \leq \frac{\min\{\mu, \frac{\gamma}{2}\} - 4L\phi}{2\ell L} \leq \frac{1}{L}$, then there exist $x = (x_k)_{k=0}^K$ close to $y = (y_k)_{k=0}^K$ (i.e. $x \in Z$) such that $x$ is under the Gradient Descent law. $\blacksquare$

**Remark 3.** *It might be interesting for the reader to realize that in the proof above we did not explicitly use the fact that $\nabla\phi(x^*) = 0$. However, we are implicitly using it by assuming gradients bounded by $\ell$: this will not be the case if we are far from a stationary point of $\phi$.*

# D Additional results

In this section we prove some results which are used in the proofs of shadowing.

## D.1 Gradient descent on strongly convex functions is contracting

> **Proposition D.1** (restated Prop. 3). *Assume (H1), (H2). For all $h \leq \frac{1}{L}$, $\Psi_h^{GD}$ is uniformly contracting with $\rho = 1 - h\mu$.*

*Proof.* The general proof idea comes from [9]. For any $x_1, x_2 \in \mathbb{R}^d$ we want to compute

$$\|\Psi_h^{GD}(x_1) - \Psi_h^{GD}(x_2)\| = \|x_1 - x_2 - h(\nabla f(x_1) - \nabla f(x_2))\|.$$

Let $w : [0,1] \to \mathbb{R}^d$ be the straight line which connects $x_2$ to $x_1$, that is $w(r) = (1-r)x_2 + rx_1$. It is clear that $w'(r) = x_1 - x_2$ and, by the fundamental theorem of calculus (FTC),

$$\nabla f(x_1) - \nabla f(x_2) = \int_0^1 \frac{d(\nabla f(w(r)))}{dr} dr = \left( \int_0^1 \nabla^2 f(w(r)) dr \right)(x_1 - x_2).$$

*It is of chief importance to notice that, in general, we can only use the FTC if $f \in C^2(\mathbb{R}^d, \mathbb{R})$:* requiring the objective to be just twice differentiable is not sufficient; indeed, if the integrand is not continuous, the integral will not in general be differentiable : as a result, the Hessian would be undefined at certain points. Next, notice that, since clearly $\bar{H} := \int_0^1 \nabla^2 f(w(r)) dr$ satisfies $\mu I \leq \bar{H} \leq LI$, we get

$$\|\Psi_h^{GD}(x_1) - \Psi_h^{GD}(x_2)\| = \|(I - h\bar{H})(x_1 - x_2)\| \leq \|I - h\bar{H}\|\|x_1 - x_2\| \leq (1 - h\mu)\|x_1 - x_2\|,$$

where we used that $\bar{H}$ is symmetric (matrix norm is the biggest eigenvalue) and $h \leq \frac{1}{L}$. ∎

## D.2 Heavy-ball local approximation error from HB-ODE

We first need a lemma.

**Lemma 1.** *Assume (H1). Let $(p, q)$ be the solution to HB-ODE starting from $p(0) = 0$ and from any $q \in \mathbb{R}^d$, for all $t \geq 0$, we have*

$$\|p(t)\| \leq \ell/\alpha,$$
$$\|\dot{p}(t)\| \leq 2\ell,$$
$$\|\ddot{p}(t)\| \leq 2(\alpha + L)\ell.$$

*Proof.* Let $S(t) = e^{\alpha t}p(t)$, then $\dot{S} = \alpha e^{\alpha t}p(t) + e^{\alpha t}(-\alpha p(t) - \nabla f(p(t))) = -e^{\alpha t}\nabla f(p(t))$. Hence, since $p(0) = 0$, $e^{\alpha t}p(t) = \int_0^t e^{\alpha s}\nabla f(p(s))ds$. Therefore

$$\|p(t)\| = \left\|e^{-\alpha t}\int_0^t e^{\alpha s}\nabla f(p(s))ds\right\| \leq e^{-\alpha t}\int_0^t e^{\alpha s}\|\nabla f(p(s))\|ds$$

$$\leq e^{-\alpha t}\int_0^t e^{\alpha s}\ell ds = \frac{1 - e^{-\alpha t}}{\alpha}\ell \leq \frac{\ell}{\alpha}.$$

Using this, we can bound the acceleration

$$\|\dot{p}(t)\| = \|-\alpha p(t) - \nabla f(p(t))\| \leq \alpha\|p(t)\| + \|\nabla f(p(t))\| \leq 2\ell$$

and the jerk

$$\|\ddot{p}(t)\| = \left\|-\alpha\dot{p}(t) - \frac{d}{dt}\nabla f(p(t))\right\| \leq 2\alpha\ell + \|\nabla^2 f(p(t))\|\|\dot{p}(t)\| = 2(\alpha + L)\ell.$$

∎

We proceed with the proof of the proposition.

**Proposition D.2** (restated Prop. 5). *Assume (H1). The discretized HB-ODE solution (starting with zero velocity) $(y_k)_{k=0}^{\infty}$ is a $\delta$-pseudo-orbit of $\Psi_{\alpha,h}^{HB}$ with $\delta = \ell(\alpha + 1 + L)h^2$:*

$$\|y_{k+1} - \Psi_{\alpha,h}^{HB}(y_k)\| \leq \delta, \quad \textit{for all } x \in \mathbb{R}^d.$$

*Proof.* Thanks to Thm. 1, since the solution $y = (p, q)$ of HB-ODE is a $C^2$ curve, we can write $(p(kh + h), q(kh + h)) = y(kh + h) = \varphi_h^{GD}(y(kh)) = y_{k+1}$ using Taylor's theorem with Lagrange's Remainder in Banach spaces (as in the proof of Thm. 2) around time $t = kh$:

$$
\begin{aligned}
p(kh + h) &= p(kh) + h\dot{p}(kh) + \mathcal{R}_p(2, h) \\
&= p(kh) + h(-\alpha p(kh) - \nabla f(q(kh))) + \mathcal{R}_p(2, h) \\
&= (1 - h\alpha)p_k - \nabla f(q_k) + \mathcal{R}_p(2, h)
\end{aligned}
\tag{7}
$$

and

$$
\begin{aligned}
q(kh) &= q(hk + h) + h\dot{q}(hk + h) + \mathcal{R}_q(2, h) \\
&= q(hk + h) + hp(kh + h) + \mathcal{R}_q(2, h).
\end{aligned}
\tag{8}
$$

where

$$\|\mathcal{R}_p(2, h)\| \leq \frac{h^2}{2} \sup_{0 \leq \lambda \leq 1} \|\ddot{p}(t + \lambda h)\| \overset{\text{Lemma 1}}{\leq} h^2(\alpha + L)\ell$$

$$\|\mathcal{R}_q(2, h)\| \leq \frac{h^2}{2} \sup_{0 \leq \lambda \leq 1} \|\ddot{q}(t + \lambda h)\| = \frac{h^2}{2} \sup_{0 \leq \lambda \leq 1} \|\dot{p}(t + \lambda h)\| \leq \| \quad \overset{\text{Lemma 1}}{\leq} \quad = h^2\ell.$$

Without the residuals, Eq. 7 and 8 are the exact equations of the integrator $\Psi_{\alpha,h}^{HB}$ (Eq. HB-PS in main paper). Hence, for all $y_k = (p_k, q_k)$ in the pseudo orbit (which we require to start at $p(0) = 0$), by the triangle inequality,

$$\|\Psi_{\alpha,h}^{HB}(p_k, q_k) - \varphi_{\alpha,h}^{HB}(p_k, q_k)\| \leq \|\mathcal{R}_q(2, h)\| + \|\mathcal{R}_p(2, h)\| \leq \ell(\alpha + 1 + L)h^2.$$

∎

### D.3  Heavy-ball on a quadratic is linear hyperbolic

We want to study hyperbolicity of the map $\Psi_{\alpha,h}^{HB}$ in phase space $(v, z)$. This map was defined in Eq. HB-PS, we report it below:

$$
\begin{cases}
v_{k+1} = v_k + h(-\alpha v_k - \nabla f(z_k)) \\
z_{k+1} = z_k + h v_{k+1}
\end{cases}
\tag{9}
$$

**Theorem D.1.** *Let $f : \mathbb{R}^d \to \mathbb{R}$ be an $L$-smooth quadratic with Hessian $H$. Assume $H$ does not have the eigenvalue zero. If $h \leq \sqrt{\frac{2}{L}}$ and $\alpha$ is such that $(1 - h\alpha) =: \beta \in [0, 1)$, then $\Psi_{\alpha,h}^{HB}$ is linear hyperbolic in phase space.*

*Proof.* If $\nabla f(x) = H(z_k - z^*)$, by adding and subtracting $z^*$ from both sides in the second equation, we can write this as a linear system

$$
\begin{pmatrix} z_{k+1} - z^* \\ v_{k+1} \end{pmatrix} = A \begin{pmatrix} z_k - z^* \\ v_k \end{pmatrix},
$$

where

$$
A = \begin{pmatrix} I - h^2 H & \beta h I \\ -hH & \beta I \end{pmatrix} \quad \text{and} \quad \beta = 1 - \alpha h.
$$

We assume $H$ has no eigenvalue at zero, and we seek to know if $A$ has eigenvalues on the unit circle $\mathbb{S}^1$. If there are no such eigenvalues, then Heavy-ball is hyperbolic.

To find the eigenvalues of this matrix we consider solving the following eigenvalue problem: for some $\xi_v, \xi_x \in \mathbb{R}^d$ and $q \in \mathbb{R}$, we require that

$$A \begin{pmatrix} \xi_x \\ \xi_v \end{pmatrix} = q \begin{pmatrix} \xi_x \\ \xi_v \end{pmatrix}.$$

To start, notice that this implies $-hHy_k = (q - \beta)\xi_v$. Therefore, assuming $q \neq \beta$, the position and velocity part of the eigenvector are linked:

$$\xi_v = \frac{hH}{\beta - q} y_k.$$

Hence, we get

$$(I - h^2 H)y_k + \frac{h^2 \beta H}{\beta - q}\xi_x = q\xi_x.$$

Therefore, we need $((\beta - q)I - (\beta - q)h^2 H + h^2 \beta H - q(\beta - q))\xi_x = 0$. Hence, $\xi_x$ needs to be an eigenvector of $H$, relative to some eigenvalue $\lambda$ which satisfies

$$\beta - q - (\beta - q)h^2\lambda + h^2\beta\lambda - q(\beta - q) = 0$$
$$\implies \beta - q - \beta h^2\lambda + qh^2\lambda + h^2\beta\lambda - \beta q + q^2 = 0,$$

which results in the simple quadratic equation

$$\boxed{q^2 - (\beta + 1 - h^2\lambda)q + \beta = 0.} \tag{10}$$

A similar equation was derived in [29]. This is the fundamental equation we have to study for hyperbolicity. We have to show that, under some assumptions on $h$, $\lambda$, $\beta$, the solution $p$ never has norm one.

Case with complex roots. If the roots of Eq. 10 are complex conjugates, $p, \bar{p}$, then

$$q^2 - (\beta + 1 - h^2\lambda)q + \beta = (q - p)(q - \bar{p}).$$

Hence, $\beta = |p|^2$. Therefore, if $\beta \in [0, 1)$, so are the moduli of the complex roots.

Case with real roots. We consider the more general equation $x^2 + bx + c = 0$. It's easy to realize that, if the roots are real, then a necessary conditions for those to lie on the unit circle is $c = -1 \pm b$. Indeed,

$$|x_{sol1}| = 1 \text{ or } |x_{sol2}| = 1 \iff |-b \pm \sqrt{b^2 - 4c}| = 2$$
$$\iff b^2 + (b^2 - 4c) \pm 2b\sqrt{b^2 - 4c} = 4$$
$$\iff b^2 - 2c - 2 = \pm b\sqrt{b^2 - 4c}$$
$$\implies b^4 + 4c^2 + 4 - 4b^2c - 4b^2 + 8c = b^2(b^2 - 4c)$$
$$\iff c^2 + 2c + 1 - b^2 = 0$$
$$\iff c = \frac{-2 \pm \sqrt{4 - 4(1 - b^2)}}{2}$$
$$\iff c = -1 \pm b.$$

in our case, $b = -\beta - 1 + h^2\lambda$ and $c = 1$. Therefore, we have to check that the following conditions are *never* fulfilled.

1. $-1 - \beta - 1 + h^2\lambda = \beta$. This implies $h^2 = \frac{\beta + 2}{\lambda}$. If $\lambda < 0$, since $\beta \in [0, 1)$, the equation is never true. Else, it is true if $h = \sqrt{\frac{\beta + 2}{\lambda}}$. But, since $|\lambda| \leq L$, if we assume $h \leq \sqrt{\frac{2}{L}}$, the equation is never satisfied.

2. $-1 - (-\beta - 1 + h^2\lambda) = \beta$. This is verified only if $\lambda = 0$, which cannot happen under our assumptions.

$\blacksquare$

## Footnotes

[13] $f(x) \to \infty$ as $\|x\| \to \infty$

[14] $\frac{x}{1+x} \leq \frac{x}{2}$ for $0 \leq x \leq 1$.

[15]This freedom in the initial condition is not reported in the statement of theorem for the sake of simplicity: there exist many possible shadowing orbits, each corresponding to a Banach Space.

[16]This is precisely why similar arguments cannot be used for the Heavy-ball method in the quadratic case (which leads to a non-symmetric $R(hH)$), as discussed at the end of Sec. 4. Indeed, for instance, the operator norm of $A = \begin{pmatrix} 0 & 1 \\ 0 & 0 \end{pmatrix}$ is 1, but $A$ has only zero eigenvalues.