[Reviews · NeurIPS 2019]

Reviewer 1



The paper presents several "shadowing" results for gradient descent and the heavy ball method, under several conditions on the objective. In short, the authors provide conditions under which a discrete approximation of an ODE defines a trajectory that "stays close" to the actual trajectory of the ODE. This research is motivated by a by a recent paper by Su, Jordan, and Candes that models Nesterov's method via an ODE: this leads the authors to ask the question of when an ODE solution indeed well approximates a discrete algorithm, which is what would be implemented in practice. Although the interest and motivation is mostly on HB, the bulk of the results presented in the paper are for GD. The paper is well-written overall, and the results are interesting, if somewhat shallow. There is also a question of practicality, as it is not clear how to use these results other than in the negative (e.g., no convergence rates seem inferable from these bounds). On a more fundamental level, my main concern has to do with novelty. The authors hint this, but discrete approximations of ODEs are a very old subject. The authors refer to papers in the fifties, but in fact discrete approximations of ODE's go back centuries (in fact, the authors analyze the so-called Euler method), and ideas presented here seem to be revisiting central notions of real analysis, numerical analysis, and control. Even on the stochastic side, bounds on corresponding approximations are also classic (see, e.g., the chapter called "The ODE Method" in Benveniste and Priouet, or the corresponding sections of Kushner and Yin cited by the authors). The classic nature of the topic gives me pause in accepting the paper in its current form. I feel that it does not treat this context appropriately: erring towards clarity and readability perhaps, the paper presents related work somewhat informally, and does not sufficiently place current results to the broader real analysis/numerical analysis/control literature. ————— Review edited after rebuttal ————— The authors are very right that they give a comprehensive, formal review of shadowing in Section 2; my comment was unfair and poorly worded, to say the least. Section 2 does indeed present “context”, and recent results, on what is known on shadowing properties at least from a control perspective. I do still believe that it would be worth giving a point of view from real analysis (convergence results of finite operators to trajectories) as well as from numerical analysis, where correct computation seems to be a central theme. I am not an expert in either disciplines, so I cannot provide more concrete feedback; my apologies for this.

Reviewer 2



This paper studies the deviation of the (discrete) trajectory of optimization algorithms from the (continuous) trajectory of their continuous-time ODE counterparts. The optimization algorithms considered are gradient descent and heavy ball method (also known as momentum method). The paper shows that, under some conditions of the objective function (strong convexity, smoothness and Lipschitz condition), trajectory of the gradient descent method with small enough step-size does not deviate too much from the trajectory of its ODE counterpart. The paper also extends the discussion to cases like strong concavity, saddle points and stochastic GD, but obtain conclusions that is somehow weaker than the one mentioned above. To some degree, these results verify the conjecture that analysis of the corresponding ODE would be helpful for analyzing the optimization algorithms, especially for the gradient descent on strongly convex objective functions. However, the result in Theorem 3, which is the main result, is not surprising to me at all, since the step-size is required to be very tiny and objective function is strongly convex (only one global minimum exists, which makes the trajectory stable). Note that in Theorem 3, it is almost always true that 2\mu\epsilon/(L\ell) << 1/L. Then the step-size h would be very tiny, compared to the common practical choice 1/L. The result in nonconvex case (Theorem 4) is also limited. It only holds for quadratic functions. For general non-convex function, this means Theorem 4 only holds for a small region/neighborhood, which can be well approximated by quadratic function. Outside of this region, nothing is known. More comments/concerns: >Theorem 3 requires that the objective function is strongly convex as well as Lipschitz. These two conditions are contradictory, since the objective function has to grow no slower than quadratic as required by strong convexity. Obviously quadratic function is not Lipschitz. Although this contradiction can be resolve by limiting the radius of the feasible parameter space, the Lipschitzness would depend on this radius. In this case, according to Theorem 3, larger feasible parameter space results in even smaller step size h.

Reviewer 3



I have read the authors rebuttal, and decided to keep my score. I believe that if the heuristic glueing approach can be made precise, this would greatly improve the paper. ======================== This is an interesting paper and addresses an important problem: namely the conditions under which discrete-time optimization algorithms can be approximated by continuous time dynamics uniformly in time. If this holds, large-time properties of the latter can be used to infer that of the former. Previous work in this direction relied mostly on constructing Lyapunov functions, but in this work the authors propose an alternative: allowing the initial condition of the continuous-time dynamics to change a little -- thereby establishing uniform error estimates for a larger class of loss landscapes. Although most theoretical results are based on well-established results from dynamical systems theory, I feel it is nevertheless important to introduce these ideas to the greater machine learning community. One perceived drawback of this work is the limited setting: most results are on quadratic functions (or perturbations of them). I have the following questions: 1. Can the results for quadratic settings be generalized, especially for the Hyperbolic case, to more general loss functions? e.g Thm C.1 C.2 2. On gluing: from the discussion from line 237, it would appear to me that if you perform local quadratic approximation of the loss function and then glue all pieces (assuming they are hyperbolic, uniformly) together you would naively get a global result. Since this result is not proved, I expect there is some difficulty in this program. I would appreciate an explanation of this difficulty.

[Author Response · NeurIPS 2019]

We thank all the reviewers for their valuable feedback. We address their individual concerns below.

<span style="color:blue">**Reviewer 1**</span>

**Novelty**   We agree there is a lot of prior classical work on numerical analysis ODEs and all references that are relevant
to us are cited in the paper. Reviewer 3 agrees on the importance of introducing these ideas to the machine learning
community. Besides, our contribution is in the *application* to optimization: in this regard, the results derived in Thm.
3,4 and C2 are novel. As suggested by the reviewer, we will revise the text to emphasize where our contribution lies.

**Context not treated appropriately**   While designing this paper we spent several months researching into the dynam-
ical systems and numerical analysis literature. After speaking directly with leading researchers in these fields, we
condensed our knowledge in a readable yet exhaustive overview of the theory of shadowing and error analysis (Sec. 2).
The same ideas are then presented again in a self-contained yet less abstract way in the main sections, directly applied
to optimization methods. Given the amount of effort from our side in trying to collect the most relevant ideas from
the literature to make them easily accessible to the community (most papers we cite work in more abstract settings,
relying on manifolds or using a somewhat outdated notation), *we would kindly ask the reviewer if he/she could be more*
*precise in his/her claim that we do not "treat the context appropriately", ideally by giving concrete examples.* To further
convince the reviewer that our discussion and understanding of the literature is complete, we invite him/her to check [5]:
a seminal paper in numerical analysis which shows how central notions in the analysis of Euler's method cannot be
applied to get useful global bounds (their Sec.1) hence one has to rely on perturbed hyperbolic splittings (their Sec.3).

**The ODE method**   On the same line, we also remind the reviewer that the "ODE method" results only holds
asymptotically and under stepsizes decreasing to zero. Instead, crucially, we consider fixed stepsizes and our bounds
hold from the very first iteration: this a completely different setting which *cannot* be captured by the ODE method.

**"it is not clear how to use these results other than in the negative"**   Please note that Thm. 3 is tight: the result
precisely describes the behavior of GD and we therefore believe this result is valuable for the community, even if that
could appear as a "negative" result (i.e. *not possible to improve*). We will add a comment on this, see note below.

Tightness of formula for $\epsilon$: first we note that the bound for $\delta$ in Prop. 3.1. cannot be improved; indeed it coincides with the well-
known *local* truncation error of Euler's method. Next, pick $f(x) = \frac{1}{2}x^2$, $x_0 = 1$ and $h = \frac{1}{L} = 1$. For $k \in \mathbb{N}$, gradients are smaller
than 1 for both GD-ODE and GD, hence $\ell = L = \mu = 1$. Our formula for the *global* shadowing radius gives $\epsilon = \frac{hL\ell}{2\mu} = 0.5$,
equal to the local error $\delta = \ell L h^2/2$ — i.e. as tight the well-established local result. In fact, GD jumps to 0 in one iteration, while
$y(t) = e^{-t}$; hence $y(1) - x_1 = 1/e \approx 0.37 < 0.5$. For smaller steps like $h = 0.1 < \frac{1}{L}$, our formula predicts $\epsilon = 0.05 = 10\delta$. In
simulation, we have maximum deviation at $k = 10$ and is $\approx 0.02 = 4\delta$— which is only 2.5 times smaller than our prediction. $\square$

<span style="color:blue">**Reviewer 2**</span>

**h too small**   Actually, our results hold for *any stepsize* $h \leq 1/L$: as can be seen from Thm. 3, each choice of $h$ will
determine a shadowing radius $\epsilon$ such that $h = 2\mu\epsilon/(L\ell)$. We agree that $\epsilon$ can be large in some cases, but this bound in
nonetheless tight (see answer to R1) and it's not possible to improve it. Indeed, *it captures a fundamental property of*
*the ODE approximation* — which is what we are after. We will update our conclusion section making a remark on this.

**Thm.4 only holds for quadratics**   We kindly point the reviewer to line 234 and App C4, where the mentioned result
is *generalized to non-quadratic saddles*. The extension to more complex landscapes is discussed qualitatively (see also
reply to R3) at line 238 and tested in the experimental section. We will emphasize these results in the revised version.

**Lipschitzness**   We addressed this in footnote 6: "[...] we can pick $\ell = L\|x_0 - x^*\|$ even tho quadratics are not Lip."

<span style="color:blue">**Reviewer 3**</span>

**More generality**   In full generality, the theory we present can be applied to a cost which, restricted to a (fixed)
subspace, is strongly convex and, restricted to the remaining directions, is strongly concave. This cost can be then also
perturbed as in Thm. C2. After an extensive literature search, we feel confident in claiming that the described setting is
aligned with what the classical literature on long time numerical error approximation (i.e. shadowing) can tackle. To
convince the reader of this fact, we invite him/her to read the introduction of [5]: a seminal paper in numerical analysis
which studies shadowing (i.e. long time approximation) *near hyperbolic saddles* (i.e. when there exists an approximate
hyperbolic splitting of $\mathbb{R}^d$). The tools in our paper, specifically the proof of Thm. C.2., are inspired by this work. We
will add a comment making clear what we said in the first sentence of this paragraph: i.e. *our results can be applied*
*outside the quadratic regime* (see also gluing below) and are in line with what is known in numerical analysis.

**Gluing**   We thank R3 for the interest in this idea, which we indeed only sketched and would like to discuss quanti-
tatively in the potential additional page. In short, the outlined simple gluing procedure is successful *if the number of*
*unstable directions is non-increasing*. In numerical analysis, this was explored in [9] (see 2nd paragraph of their intro).
We will add this result in the form of a theorem and make clear what objectives can be studied with this approach.

[Meta-Review · NeurIPS 2019]

The paper presents a theoretical analysis of how well a discrete dynamic flow approximates the flow/solution of a corresponding ODE for gradient descent and heavy ball methods, e.g., how trajectory of the discrete method with small enough step-size does not deviate too much from the trajectory of the ODE. The main theoretical results are somewhat limited, i.e., small step size and quadratic functinos, but are of interest.